

# An introduction to the Australian and New Zealand flux tower network – OzFlux*

**Jason Beringer[1]**, Lindsay B. Hutley[2], Ian McHugh[3], Stefan K. Arndt[4], David Campbell[5], Helen A. Cleugh[6], James Cleverly[7], Víctor Resco de Dios[8], Derek Eamus[7], Bradley Evans[9,10], Cacilia Ewenz[11], Peter Grace[12], Anne Griebel[4], Vanessa Haverd[6], Nina Hinko-Najera[4], Alfredo Huete[13], Peter Isaac[6], Kasturi Kanniah[14], Ray Leuning[6,#], Michael J. Liddell[15], Craig Macfarlane[16], Wayne Meyer[17], Caitlin Moore[3], Elise Pendall[18], Alison Phillips[19], Rebecca L. Phillips[20], Suzanne Prober[16], Natalia Restrepo-Coupe[13], Susanna Rutledge[6], Ivan Schroder[21], Richard Silberstein[22], Patricia Southall[23], Mei Sun[24], Nigel J. Tapper[3], Eva van Gorsel[6], Camilla Vote[25], Jeff Walker[24] and Tim Wardlaw[19].

[1]School of Earth and Environment (SEE), The University of Western Australia, Crawley, WA, 6009, Australia.
[2]School of Environment, Research Institute for the Environment and Livelihoods, Charles Darwin University, NT, Australia 0909.
[3]School of Earth, Atmosphere and Environment, Monash University, Clayton, 3800, Australia.
[4] School of Ecosystem and Forest Sciences, The University of Melbourne, Richmond, 3121, Victoria, Australia.
[5] School of Science, University of Waikato, Hamilton 3240, New Zealand.
[6] CSIRO Oceans & Atmosphere Flagship, Yarralumla, ACT, 2600, Australia.
[7] School of Life Sciences, University of Technology Sydney, Broadway, NSW, 2007, Australia.
[8] Producció Vegetal i Ciència Forestal, Universitat de Lleida, 25198, Leida, Spain.
[9] School of Life and Environmental Sciences, The University of Sydney, Sydney, NSW,  2006, Australia.
[10] Ecosystem Modelling and Scaling Infrastructure, Terrestrial Ecosystem Research Network, The University of Sydney, NSW, 2006.
[11] Airborne Research Australia, Flinders University, Salisbury South, SA, 5106, Australia.
[12] Institute for Future Environments and Science and Engineering Faculty, Queensland University of Technology, Brisbane, QLD 4000, Australia.
[13] Remote Sensing Research Group, Plant Functional Biology and Climate Change Cluster (C3), University of Technology Sydney, Broadway, NSW, 2007, Australia.
[14] Faculty of Geoinformation and Real Estate, Universiti Teknologi Malaysia, Johor Bahru, Johor, 81310, Malaysia.

[15] Centre for Tropical Environmental and Sustainability Science, James Cook University, Cairns, QLD, 4878, Australia.
[16] CSIRO Land and Water, Private Bag 5, Floreat 6913, Western Australia.
[17] Environment Institute, The University of Adelaide, Adelaide SA 5005, Australia.
[18] Hawkesbury Institute for the Environment, Western Sydney University, Penrith, NSW 2751 Australia.
[19] Forestry Tasmania, Hobart, Tasmania, 7000, Australia.
[20] Landcare Research, Lincoln, New Zealand.
[21] International CCS & CO2CRC, Resources Division, Geoscience Australia, Canberra, ACT, 2601, Australia.



[22] Centre for Ecosystem Management, Edith Cowan University, School of Natural Sciences, Joondalup, WA, 6027, Australia
[23] Centre for Ecosystem Management, Edith Cowan University, School of Natural Sciences, Joondalup, WA, 6027, Australia.
[24] Department of Civil Engineering, Monash University, Clayton, 3800, Australia.
[25] Graham Centre for Agricultural Innovation, Charles Sturt University, Wagga Wagga, NSW, 2678, Australia
[#] deceased

*Corresponding to: Jason Beringer, tel. +61 409355496, e-mail: Jason.Beringer@uwa.edu.au*

**Abstract.** OzFlux is the regional Australian and New Zealand flux tower network that aims to provide a continental-scale national research facility to monitor and assess trends, and improve predictions, of Australia's terrestrial biosphere and climate. This paper describes the evolution, design and current status of OzFlux as well as an overview of data processing. We analyse

measurements from the Australian portion of the OzFlux network and found that the response of Australian biomes to climate was largely consistent with global studies but that Australian systems had a lower ecosystem water-use efficiency. Australian semi-arid/arid ecosystems are important because of their huge extent (70%) and they have evolved with common moisture limitations. We also found that Australian ecosystems had similar radiation use efficiency per unit leaf area compared to global values that indicates a convergence toward a similar biochemical efficiency. The paper discusses the utility of the flux

data and the synergies between flux, remote sensing and modelling. Lastly, the paper looks ahead at the future direction of the network and concludes that there has been a substantial contribution by OzFlux and considerable opportunities remain to further advance our understanding of ecosystem response to disturbances including drought, fire, land use and land cover change, land management and climate change that are relevant both nationally and internationally. It is suggested that a synergistic approach is required to address all of the spatial, ecological, human and cultural challenges of managing the

delicately balanced ecosystems in Australia.

**Keywords.** Eddy covariance, flux tower networks, OzFlux, FLUXNET, remote sensing, modelling, climate drivers, water use efficiency, carbon cycle.

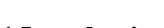

# 1 Introduction

*This paper is dedicated to the memory of our wonderful colleague and guiding light, Dr. Ray Leuning, who recently passed away.*

## 1.1 The role of flux research in Australia

Global environmental change is one of the greatest challenges facing the planet (Steffen et al., 2011). To mitigate or adapt to global environmental change we must provide a scientific basis, underpinned by observation that is then scaled using

models, for the development of national and global policies for improved land management (Vargas, 2002). Natural terrestrial ecosystems provide a range of services such as carbon sequestration and climate regulation, water balance, biodiversity, ecotourism, resources and food (Costanza et al., 1998; Eamus et al., 2005), yet they are at risk from climate change and variability, land use change and disturbance (Schroter et al., 2005). Natural ecosystems are also important sinks and sources of greenhouse gases that are sensitive to climate variability and can feed back to global climate change (Luo, 2007). Finally,

changes in physical land surface properties can occur through land use change, disturbance and biogeographical shifts in ecosystems (Burrows et al., 2014) that can, in turn, alter biophysical coupling and feedback to alter weather and climate patterns at multiple scales (Beringer et al., 2014; Bonan, 2008).

Future climate and land use change may push ecosystems towards tipping points (Laurance et al., 2011), with deleterious changes to vegetation structure, composition and function, thus compromising ecosystem health and viability

(Hughes, 2010). Of particular concern are potential increases in the effects of climate-induced physiological stress and interactions with other climate-mediated processes such as insect outbreaks and wildfire (Allen et al., 2010; Evans et al., 2013). This has important effects on carbon sequestration and greenhouse gases emissions because carbon stored in woody vegetation is vulnerable to increased fire risk through burning under climate change (Bowman et al., 2013). Death of vegetation from drought stress (Mitchell et al., 2014), extreme disturbance events, disease and pests could also result in

increased carbon release to the atmosphere and changes to $CO_2$ emissions from soils (Hutley and Beringer, 2011). It is imperative that we understand the value and dynamics of ecosystem structure and function to ensure that we can manage them successfully into the future under environmental change (fire, pests and invasive species, future land-use and climate change) (Beringer et al., 2014; Hutley and Beringer, 2011).

To address global concern over rising atmospheric $CO_2$ concentrations and global climate change, there has been a

growing need for studies of terrestrial ecosystems (Peters and Loescher, 2014). Studies using the eddy covariance technique

from micrometeorological flux towers (Baldocchi, 2003) can contribute significantly to our understanding of ecological,

biogeochemical, and hydrological processes by, amongst others:

1) providing accurate, continuous half-hourly to annual estimates of sinks and sources of greenhouse gases and water

from ecosystems for carbon accounting and water management (Eugster and Merbold, 2015; Hutley et al., 2005);

2) evaluating the effects of disturbance, topography, biodiversity, stand age, land use, insect/pathogen infestation and

extreme weather on carbon and water fluxes (Baldocchi, 2008);

3) examining the effects of land management practices, such as harvest, fertilisation, irrigation, tillage, thinning,

cultivation and clearing (Bristow et al., 2016; Prescher et al., 2010; Vote et al., 2016); and

4) producing important ground-truth data for parameterising, validating, and improving satellite remote sensing and

global inversion products (Running et al., 1999).

Direct measurements from diverse biomes are essential for developing bio-geochemical and ecological models that

diagnose and forecast the state of the land's carbon and water budgets (Baldocchi, 2014b; Haverd et al., 2013a), which

ultimately allow us to better respond and adapt to environmental change (Steffen et al., 2011). Given the utility of eddy

covariance studies and the demand for these types of data, global and regional networks have come together to maximise their

scientific value. The international network FLUXNET is a "network of regional networks" that coordinates regional and global

analysis of observations from micrometeorological flux tower sites (Baldocchi et al., 2001), where at present over 650 sites

are operated on a long-term and continuous basis. Biomes in FLUXNET include temperate conifer and broadleaved (deciduous

and evergreen) forests, tropical and boreal woodlands and forests, crops, grasslands, chaparral, arid woodlands and scrublands,

wetlands and tundra. Within FLUXNET are a number of regional networks such as AmeriFlux (USA), AsiaFlux (Asia),

ChinaFlux, Fluxnet Canada and OzFlux (Australia and New Zealand). The aim of this paper is to describe the evolution,

design and current status of the Australian network of eddy covariance flux towers (OzFlux). Although New Zealand flux sites

have been an integral part of OzFlux from the outset and have made many important contributions (Hunt et al., 2002; Rutledge

et al., 2010, 2015; Thompson et al., 1999), these sites have had a different history and data availability, thus this paper will

focus on Australian sites. An overview of data processing will be given first, followed by a summary and analysis of

measurements from the entire Australian OzFlux network. This is followed by an examination of synergies between flux

measurements, remote sensing and modelling. Australia is a vast continent and up-scaling using validated terrestrial biosphere

modelling and remote sensing products is essential for complementing limited ground based biophysical observations. We will

conclude by looking ahead at the future direction of the network.

**1.2 Evolution of OzFlux in Australia**

Australia has a long and rich history of significant contributions to the field of micrometeorology, including the

development of theory around turbulence in and above plant canopies (Deacon, 1959; Priestley, 1967; Raupach and Thom,

1981; Webb et al., 1980) and instrumentation (Deacon and Samuel, 1957; Raupach, 1978; Taylor and Dyer, 1958) along with

some of the first field measurements (Denmead and McIlroy, 1970; Hicks and Martin, 1972). Initial micrometeorological field

measurements were designed to validate the methodology and were often conducted in short campaigns over agricultural

landscapes (Leuning et al., 2004). Micrometeorological measurements for research purposes accelerated in the 1990s with

studies such as the Maritime Continent Thunderstorm Experiment (MCTEX) (Beringer and Tapper, 2002) and OASIS (Isaac

et al., 2004). Long-term flux measurements using the eddy covariance technique began in Australia in 2000 with the

establishment of the wet temperate eucalypt forest site at Tumbarumba (Leuning et al., 2005), closely followed by the tropical

savanna site at Howard Springs (Hutley et al., 2000) and the tropical rainforest site at Cape Tribulation in 2001 (Liddell et al.,

2016) (Fig. 1). Almost all of the sites in the OzFlux network have been initially established under short term research grants

for specific purposes. However, due to the vision of the investigators who recognised the importance of long-term

measurements, these sites were kept operational on minimal budgets, which has provided a legacy of important flux and

ancillary data. At about this time the 'OzFlux' network was founded by Ray Leuning and colleagues at an inaugural meeting

at Monash University in 2001 (Leuning et al., 2001). Leuning was the pioneer and leader of the network for many years and

a mentor of many Australian and internationally based micrometeorologists and ecophysiologists (Cleugh, 2013). We dedicate

this paper to him.

After the establishment of OzFlux, the community lobbied the Federal Government to allocate financial resources for an ecological observational network. Over the next decade, the national collaborative research infrastructure strategy (NCRIS)

established the terrestrial ecosystem research network (TERN, 2016) as a crucial platform to integrate datasets collected by different state agencies, CSIRO (Commonwealth Scientific and Industrial Research Organisation) and Universities for supporting decision-making to overcome Australia's developing environmental problems (State of the Environment 2011 Committee, 2011).  In 2009 initial funding was provided to TERN, which provided nominal support for many OzFlux sites along with other capabilities such as intensive ecosystem monitoring (SuperSites), remote sensing (AusCover), modelling

(eMAST), TERN synthesis (ACEAS), coastal, soils and plot based networks (AusPlots), long term ecological research network facilities (LTERN) and transects (Australian Transect Network).  OzFlux has had a central network capacity and from the outset this has been hosted by the CSIRO with data services provided at present through the NCI (Australia's National Computational Infrastructure). Despite the critical information provided by TERN and OzFlux Networks, recent funding and programmatic cuts may compromise sustained environmental research in Australia.


## 2 OzFlux network architecture

### 2.1 Network overview

OzFlux aims to provide a national research facility for monitoring and assessing trends, and to improve predictions of Australia's terrestrial biosphere and climate.  It underpins the data collection and process understanding needed to: 1) support

sound management of natural resources including water, carbon and nutrient resources for environmental and production benefits; 2) monitor, assess, predict and respond to climate change and variability; 3) improve weather and environmental information and prediction; 4) support disaster management and early warning systems needed to meet Australia's priorities in national security; and 5) ensure that Earth system models used to underpin Australia's policies and commitments to international treaties adequately represent Australian terrestrial ecosystem processes.

The key ecosystem science questions are focused on improved understanding of the responses of carbon and water cycles of Australian ecosystems to current climate and future changes in precipitation, temperature and $CO_2$ levels, as follows; 1)

what are the key drivers of ecosystem productivity (carbon sinks) and greenhouse gas emissions; 2) how resilient is ecosystem productivity to a variable and changing climate and 3) what is the current water budget of the dominant Australian ecosystems and how will it change in the future?


## 2.2 Network design

OzFlux established an agreed set of core measurements and common protocols for measurements of carbon, water and energy fluxes across the national network to provide consistent observations to serve the land surface and ecosystem modelling communities. In addition to long-term fluxes of carbon, water and energy, ecosystem structural and functional

properties are being measured, along with biodiversity and soil characteristics in collaboration with the TERN SuperSite Network (Australian SuperSite Network, 2015). The OzFlux network design is based on a hub and spoke model, wherein a critical element is the central node that coordinates the network¸ proposes protocols for measurements, has oversight of data processing and quality control, maintains a database to archive data from each site, data licencing and access via an online portal and provides scientific and technical training to flux station operators (Isaac et al., 2016). The central node implements

a centralised database and provides feedback on live data feeds (equipment failure) and measurement quality to site operators. Site operators have responsibility for operation, data processing, data quality control, gap filling and delivery of data streams to the central database. Through annual meetings, training workshops and technical support, OzFlux also provides a critical capacity-building opportunity to further build Australian expertise, infrastructure, and data processing across the broader Australian research community in ecosystem and climate sciences.


## 2.3 Network climate and biome space

The modified Köppen climate classification of Stern and Dahni (2013) shows that the greater part of the Australian continent is either desert climate (i.e. arid) (38% of land) or grassland climate (i.e. semi-arid) (36% of land). Only the south-east and south-west corners have a temperate climate (10%) and moderately fertile soil (McKenzie et al., 2004). The northern





third of the continent is dominated by sub-tropical (7%), tropical (9%) and grassland (36%) climates, with tropical rainforests,

tropical savanna, grasslands and deserts the dominant ecosystems.  Mean annual precipitation (MAP) across the continent

varies from 134 to 2804 mm, and mean annual temperature (MAT) varies from 3.8 to 29.0 ºC (from 1961-1990 MAP and

MAT gridded data at 0.1 degree resolution (Bureau-of-Meteorology, 2013)).  Many individual stations exceed the spatially

gridded data due to topographical/spatial issues such as Mount Bellenden Ker that has the highest mean annual rainfall of any

Bureau of Meteorology station at 8173 mm (station ID 031141 for period 1973 to 2015).  The OzFlux sites are relatively

uniformly distributed over this climate space (Fig. 2).

Using the modified Köppen scheme, Stern and Dahni (2013) showed that the major Australian climate zones were

(from largest to smallest by area) desert, grassland, temperate, tropical, sub-tropical, equatorial and polar.  Stern and Dahni

(2013) also reported changes in the distribution of Australian climate zones due to climate change during the past century, the

most notable one being the contraction of the area covered by 'desert' climates (from 51.1% to 37.9%) and the corresponding

increase in the area covered by 'grassland' (from 26.3% to 36.1%) and 'tropical' (from 5.5% to 9.0%) climates.  Therefore, a

large portion of the continent is arid or semi-arid (74%) and ecosystems in the semi-arid climate zone have been recognized

recently as of critical important in driving inter-annual variability in the global $CO_2$ growth rate cycle (Bastos et al., 2013;

Poulter et al., 2014).  Poulter et al. (2014), Haverd et al. (2015) and Cleverly et al. (2016a) found that a large sink anomaly in

2011 was mainly attributed to increases in net primary productivity (NPP) across the semi-arid regions of the southern

hemisphere (30-60% from Australian semi-arid ecosystems), during a large La Niña event where Australian MAP exceeded

the long-term by 55% (Boening et al., 2012).  As a consequence the water-limited ecosystems responded by rapid growth and

productivity.  Ahlström et al. (2015) subsequently demonstrated that both the inter-annual variability and trend of the global

sink were dominated by semi-arid ecosystems whose carbon balance is strongly associated with circulation-driven variations

in both precipitation and temperature.  This and similar dynamics have been captured at the semi-arid sites in OzFlux as

discussed in Eamus and Cleverly (2015) and Cleverly et al. (2016a).

In terms of vegetation classification, we have used the Interim Biogeographic Regionalisation for Australia v. 7

(IBRA) (Environment, 2012) throughout (Table 1, Fig. 1) to describe the Australian vegetation types (bioregions) and

ecoregions (global classification). Flux sites located in natural vegetation (n=27) (Table 1) cover a wide geographical and

biome space, although each ecoregion is not equally represented (Table 2, Fig. 1). Despite the dominance of deserts and xeric

shrublands (49%) and the areal importance of arid/semi-arid climate (74%), only a small fraction of the OzFlux sites (two

towers, 8% of the network) are located in this region. There is a strong representation of flux towers in the tropical and sub-

tropical moist broadleaf forests. In addition, of the 32 flux tower sites in Table 1, only five (16%) are located in predominantly

agricultural/managed/modified landscapes. Vote et al. (2016) give a comparative analysis of flux observations in Australian

managed landscapes of varying vegetation structure, composition and management intensities in different bioclimatic regions

and Bristow et al. (2016) provide a specific case study of land use transitions in tropical savannas.

Of the 11 currently TERN funded OzFlux sites 10 are also TERN SuperSites that carry out a standard set of

measurements using agreed protocols that provide a considerable set of ancillary measurements available at each OzFlux site

(http://www.supersites.net.au ). Included in this suite of data are soil characterisation, plant biodiversity, leaf area index (LAI),

vegetation structure, groundwater data, stream chemistry and faunal biodiversity (Karan et al., 2016). In addition, each

SuperSite has been supported by the TERN Auscover remote sensing facility with ground based (terrestrial laser scanner) and

air-borne (lidar, hyperspectral) remote sensing data collected in a 5 km x 5 km pixel centered at each tower. Like all TERN

data these data sets are publically available from the TERN data portals with metadata made available across the portals at

TERN (http://portal.tern.org.au ).


## 3 Eddy covariance data

### 3.1 Instrumentation and data collection

In 2016 the OzFlux network comprised 23 active flux towers across Australia (Fig. 2, Table 1). There is a high degree

of consistency in instrumentation across the OzFlux network. The general tower configuration consists of a CSAT3 sonic

anemometer (Campbell Scientific, Logan, Utah, USA) and a Li-7500[A] (LI-COR, Lincoln, Nebraska, USA) or EC-150/155

(Campbell Scientific, Logan, Utah, USA) infra-red gas analyser mounted at the top of the tower. All sites record three

components of the wind field, air temperature and the $H_2O$ and $CO_2$ concentrations at 10 or 20 Hz. Complementary

measurements of slow response wind speed (Gill Instruments Ltd, Lymington, Hampshire, UK; R.M. Young, Traverse City,

Michigan, USA), air temperature and humidity (various, Vaisala, Helsinki, Finland) are also made at least one height. Soil

water content (various, Campbell Scientific, Logan, Utah, USA), soil temperature (TCAV, Campbell Scientific, Logan, Utah,

USA) and ground heat flux (CN3, Middleton, Newtown, VIC, Australia; HFT3, Campbell Scientific, Logan, Utah, USA;

HFP01, Hukseflux, Delft, The Netherlands) are measured in soil pits adjacent to the towers and often replicated in space and

depth. Radiation (4-component) is measured at the tower top (CNR1, CNR4 Kipp & Zonen; NR01 Hukseflux; Delft,

Netherlands). Precipitation (CS702, Campbell Scientific, Logan Utah, USA; CS7000, Hydrological Services, Warwick Farm,

NSW, Australia) is measured at ground level at most sites. Systems measuring the profiles of $H_2O$ and $CO_2$ concentration in

the canopy are installed at Tumbarumba, Wombat Forest, Cumberland Plains, Whroo and Robson Creek. Details of the

instrumentation at each site are available from the OzFlux web site (http://www.ozflux.org.au/monitoringsites/index.html).

Data are recorded at most sites by Campbell data loggers (various, Campbell Scientific, Logan, Utah, USA).

Tumbarumba, Otway and Virginia Park use purpose-built micro-computers. At all sites using Campbell data loggers, the

averaged and high frequency data are retrieved via modem or recorded on compact flash (CF) cards, which are retrieved

periodically, read and archived at the site PI's institute.

### 3.2 Data quality control and post-processing

Most sites with Campbell data loggers begin with the average (over 30 minutes) covariances recorded by the logger

and processed through six levels using the OzFluxQC standard software processing scripts. For details see Isaac et al. (2016)

but in brief, levels 1, 2 and 3 represent the raw data as received from the flux tower (L1), quality-controlled data (L2) and post-

processed, corrected but not gap-filled data (L3). OzFlux sites submit their data to FLUXNET at L3. Levels 4, 5 and 6

represent data with gap-filled meteorology (L4), gap-filled fluxes (L5) and partitioned into gross primary production (GPP)

and ecosystem respiration (ER) (L6). The L1 to L3 data used in this paper have been produced using OzFluxQC (Isaac et al.,

2016) and Level 3 are then gap filled and partitioned using the Dynamic INtegrated Gap filling and partitioning for Ozflux

(DINGO) system developed by Beringer as described in Donohue et al. (2014).

OzFluxQC quality control measures are applied at L2 and include checks for plausible value ranges, spike detection

and removal, manual exclusion of date and time ranges and diagnostic checks for all quantities involved in the calculations to

correct the fluxes. The quality checks make use of the diagnostic information provided by the sonic anemometer and the infra-

red gas analyser. For sites calculating fluxes from the averaged covariances, post-processing includes 2-dimensional

coordinate rotation, low- and high-pass frequency correction, conversion of virtual heat flux to sensible heat flux and

application of the WPL correction to the latent heat and $CO_2$ fluxes (see Burba (2013) for general description of the data

processing pathways). Steps performed at L3 include the correction of the ground heat flux for storage in the layer above the

heat flux plates (Mayocchi and Bristow, 1995) and correction of the $CO_2$ flux data for storage in the canopy (where available).

OzFlux data are available at http://data.ozflux.org.au/.

## 4 Biotic and abiotic controls on land-surface exchanges

We used the conceptual framework for carbon balance terms following Chapin et al. (2006), including their sign

convention where net and gross carbon uptake (net ecosystem production (NEP) and gross primary production (GPP)) are

positive directed toward the surface and ecosystem respiration (ER) is positive directed away from the surface. For ease, the

following analysis has been aggregated by ecoregion (Table 2) from the individual site data that are detailed in Table 3. Note

that the number of years representing each site is different. Net ecosystem production (NEP) across the ecoregions varied with

forests generally the strongest sink (followed by grasslands, savannas and shrublands). The Mediterranean ecosystems were

close to carbon neutral and deserts and xeric shrublands were a carbon source overall (Table 3). This can be compared to

AsiaFlux, where NEP varied across the network between -2 and 8 tC ha$^{-1}$ yr$^{-1}$ with the differences due mainly to tree species

and mean annual temperature (MAT) (Yamamoto et al., 2005). Across all of the OzFlux sites the average NEP was 1.8 tC ha$^{-1}$

yr$^{-1}$ which is comparable to the global average of 1.6 tC ha$^{-1}$ yr$^{-1}$ (Baldocchi, 2008).

There was also a large variation in the seasonality of carbon fluxes (Fig. 3) between the ecoregions, which followed

patterns in temperature and/or rainfall and moisture availability across the continent. Tropical moist broadleaf forests had the

smallest seasonality followed by savannas, which are seasonally water limited with seasonal variation driven by large dry

season decline in fluxes from the $C_4$ grass-dominated (Whitley et al., 2011). Tropical grasslands showed the largest seasonality

and notably had the largest peak NEP of 4.5 gC m$^{-2}$ d$^{-1}$ which occurs after re-sprouting and green-up during rapid growth

when soil respiration remains low (Fig. 3a). Only later during the season, when the grasses senesce is carbon returned to the atmosphere via fire and respiration and to the soil through litter and root carbon. Across the ecoregions, GPP generally scaled

with leaf area index (LAI) and water availability (precipitation) (Fig. 5), with desert shrublands having the lowest GPP and tropical moist broadleaf forests having the highest (Table 2) (Fig. 3b). Tropical grassland GPP was similar to that of forests during the monsoonal summer wet season; however, GPP in the grassland collapsed to near zero during the dry season (Fig. 3b). In general, the magnitude of ecosystem repiration (ER) was near a constant fraction of GPP across the annual cycle for each ecoregion (Fig. 3c and Fig. 4).

290        Following on from the average fluxes (discussed above), there are periods when substantial inter-annual variability is superimposed on Australia's mean climate (Cleverly et al., 2016b) and this variability has been captured by bush poet Dorothea Mackellar as the land of "droughts and flooding rains" (Mackellar, 2011). Australia's weather is primarily driven by three climate modes: El Niño-Southern Oscillation (ENSO), the Indian Ocean dipole (IOD) and the southern annular mode (SAM) and when these climate modes synchronise, fluctuations between drought and extreme precipitation can be extreme

and rapid (Cleverly et al., 2016b). Extreme weather across Australia during the 21st century has been the result of synchronisation amongst these climate modes, such that wet conditions created by one climate mode were reinforced by similarly wet conditions in the other modes (Cleverly et al., 2016b). The El Niño phase brings warmer and drier than average conditions, whilst cooler and wetter conditions are characteristic of La Niña phase (Nicholls et al., 1991; Power et al., 1999) and together all three modes of variability influence patterns of vegetation fluxes (Cleverly et al., 2016b). In general, rainfall

is limiting to productivity across much of Australia whereas temperature is not. Raupach et al. (2013), through a modelling study, showed that evapotranspiration (ET) from Australian ecosystems is expected to increase with increasing precipitation and temperature, but decrease with rising $CO_2$ through increased plant water-use-efficiency. They also showed that NEP is expected to increase with rising $CO_2$ concentration but this may be offset by reduced NEP in response to warming. Much of the network is, either directly or indirectly, contributing critical data to refine our understanding of the drivers of NEP and the

role of precipitation events on carbon and water cycles (Chen et al., 2014; Eamus et al., 2013a; Kanniah et al., 2011; Ma et al., 2013a).

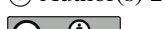



Climate variability and land management also drives recurrent fire on the Australian landscape, with Australia being one of the most fire-prone continents on earth (Bradstock et al., 2012). Using AVHRR satellite data from 1997-2005, Russell-Smith et al. (2007) showed that the distribution of large fires varied with biophysical variables, and continental fire patterns

varied substantially with rainfall seasonality. Their results highlight the importance of anthropogenic ignition sources, especially in the northern wet–dry tropics and demi-arid/arid Australia. These recent patterns differ greatly from assumed fire regimes  under Indigenous occupancy, and the differences in fire regime can cause significant effects on biodiversity that are likely to increase in the future (Russell-Smith et al., 2007). Interestingly, a number of Australian flux sites have been influenced by wildfire including: 1) a catastrophic, stand replacing wildfire in the old growth Mountain ash forests (*Eucalyptus*

*regnans,* Au-Wac) during February 2009; 2) a fire at Calperum (Au-Cpr) in January 2014 that burned spinifex (*Triodia* sp) ground cover and leaves and branches of eucalypt species; 3) frequent and relatively low intensity fires across all the savanna sites (Au-How, Au-Ade, Au-DaS, Au-Dry) and at a tropical grassland (Au-Stp). Many of the research questions, particularly in savannas, are focused on the influence of burning at scales from leaf to landscape (Beringer et al., 2003, 2007, 2014).

Annual NEP as measured by flux towers is the difference between GPP and ER (Chapin et al., 2006). On a site-by-

site basis across the network, ER and GPP are strongly correlated ($r^2$=0.93) (Fig. 4) (Reichstein et al., 2005). The slope of the line is 0.76, which compares well with an international synthesis that found a slope of 0.77 (Baldocchi, 2008). The results from the international study are shown in the background of Fig. 4, where disturbed sites fall along a secondary line (carbon source). Interestingly, few Australian sites are close to this disturbed level, except for Au-RDF, which had undergone a transition from savanna to pasture, and Au-Wrr, which is a tall forest with potential advection issues such that the ratio of GPP

to ER might be unreliable. Despite this, there are many flux towers in the network that have captured varying levels of disturbance including fire, insect attack (van Gorsel et al., 2013), logging , grazing, termite herbivory (Jamali et al., 2011) and tropical cyclones (Hutley et al., 2013). Curiously, as pointed out by Baldocchi (2008), Australian systems that are burnt frequently (i.e. savannas) do not fall in line with other types of disturbance (Fig. 4) because these low intensity fires form a type of rapid respiration that does not significantly alter carbon pools on annual timescales, either for vegetation (Beringer et

al., 2003, 2007) or as soil carbon (Allen et al. 2014).

We used a simple heat map to identify correlations between fluxes, driving variables and functional attributes of Australian ecosystems (Fig. 5). We used the following ecosystem indices: radiation use efficiency (RUE) following Garbulsky et al. (2010); ecosystem water-use-efficiency (WUE* calculated as the ratio of GPP to AET, where * indicates ecosystem scale) and inherent ecosystem WUE* (IWUE* calculated as the ratio of GPP to AET x VPD) following Beer et al. (2009);

Bowen ratio (BR) is defined as the ratio between sensible and latent heat fluxes (Bowen, 1926); leaf area index (LAI) obtained from the average of the MODIS MOD15 product for the site years available and has been de-spiked using the procedure described in Kanniah et al. (2009a); and rainfall use efficiency (Rain Use Eff) following Huxman et al. (2004) and defined as the ratio of GPP to precipitation. Note that the magnitude of LAI from the MODIS LAI product utilised in this paper varies from site based estimates but has been used for consistency.

In general, the major controls on site-averaged GPP and NEP were precipitation, vapour pressure deficit (VPD) and LAI (Fig. 5) as expected (Yi et al., 2010). Counterintuitively, incoming solar radiation ($F_{sd}$) was negatively correlated with GPP, suggesting that $F_{sd}$ is not limiting; we speculate that the negative correlation is explained by an association between regions of high sunlight ($F_{sd}$) in arid-semi/arid climates that have vegetation that tends to have lower GPP due to water limitations. Given LAI is such an important driver (and LAI is strongly correlated with precipitation), we normalised the

fluxes and indices by dividing them by LAI. Subsequently, these normalised ratios showed that after accounting for LAI, GPP was only weakly positively correlated (r<0.3) with temperature ($T_a$) and negatively correlated with precipitation (Fig. 5). We hypothesise that the negative correlation of GPP/LAI with rainfall is due to the lower radiation due to cloud associated with high rainfall.

We further explored the relationships of GPP *versus* mean annual temperature (MAT) and mean annual precipitation

(MAP). We followed Garbulsky et al. (2010) to allow for a direct comparison with that global study where they used 35 eddy covariance (EC) flux sites (none from Australia) spanning between 100 and 2200 mm MAP and between -13 and 26°C MAT. The global relationships are shown in the background to aid comparison (Fig. 6). The range of GPP across the Australian sites was 32 to 2370 gC m$^{-2}$ yr$^{-1}$ which is consistent with the range of global values (50 to 3250 gC m$^{-2}$ yr$^{-1}$) (Fig. 6). The relationship between GPP and MAP follows the global function (Fig. 6a) although the range of MAP across the Australian sites is larger

than is observed from the global flux network due to the high MAP of the sites in the tropical rainforest region. In contrast to



the global study, MAT had no significant relationship with GPP (Fig. 6b) in Australia. This is partly because there are no sites in Australia with MAT below zero and MAT is generally not limiting to GPP, however, high temperature can limit NEP in the desert ecoregion (Cleverly et al., 2013, 2016a). The most highly productive sites were obviously not moisture limited, including cool temperate forests (MAT < ~12°C) and hot wet tropical forests (MAT > ~25 °C, MAP > 2000 mm) (Fig. 6a, b). The peak

in GPP seen at MAT ~12 °C is consistent with an analysis of plot data from Australian temperate forests by Bowman et al. (2014) who noted maximum growth occurring at a mean annual temperature of 11 °C and maximum temperature of the warmest month of 25–27 °C. They found that lower temperatures directly constrained growth whilst high temperatures primarily reduced growth by reducing water availability.

Radiation-use-efficiency (RUE) across Australian ecoregions was tightly coupled with GPP and was similar (but

perhaps higher) than the international relationship (Fig. 6d). There is a similarly large range in RUE across Australian ecoregions, from pasture (0.65 gC MJ PAR$^{-1}$), temperate and Mediterranean woodland (0.75 gC MJ PAR$^{-1}$), tropical savanna (1.0 gC MJ PAR$^{-1}$) and temperate and tropical forest (1.5 gC MJ PAR$^{-1}$). Global values across FluxNet showed RUE to be approximately 0.2 gC MJ PAR$^{-1}$ and 0.35 gC MJ PAR$^{-1}$ for grassland and savanna respectively (Reichstein et al., 2014). The correlation between RUE and GPP decreased when GPP was expressed per unit LAI (i.e. GPP / LAI – see Fig. 5) suggesting

that all Australian ecosystems have similar efficiency per unit leaf and converge toward a similar biochemical efficiency.

The ecosystem water-use-efficiency (WUE*) of Australian systems was systematically lower than the global relationship (Fig. 6c) suggesting that these ecosystems have a low C gain per unit water loss. This surprising result is likely to reflect high soil evaporation ratios from open canopied woodlands and shrublands of the arid/semi-arid regions that dominate Australia. Haverd et al. (2013a) showed in BIOS2 modelling that over half (64%) of Australian ET is attributable to soil

evaporation, which is much higher than the global fraction of 27%, that results in higher water loss per unit of C gained at the ecosystem scale. There are other important factors that could explain the smaller ecosystem WUE* of the non-arid Australian systems including: 1) it may reflect reduced leaf-scale instantaneous transpiration efficiency (ITE=A/E) (Barton et al., 2012); 2) the high degree of sclerophylly of Australian vegetation whereby C assimilation rates per unit leaf area will be low (compared to the USA or European flora) because of the large investment in thick cell walls and defensive compounds

characteristic of long-lived sclerophyllous leaves (Eamus and Prichard, 1998); 3) leaf-level ITE of deciduous species is

generally larger than that of evergreen species (Eamus and Prichard, 1998; Medina and Francisco, 1994) and the flora of US and European broadleaf forests are almost exclusively deciduous; 4) optimality theory of stomatal behaviour predicts that ITE is inversely proportional to VPD (Medlyn et al., 2011) and since mean VPD is generally larger in Australia than the US and Europe, a smaller ITE is expected for Australian sites compared to global analyses that omit Australian sites and 5) low soil

nutrient availability of Australian soils (McKenzie et al., 2004) that limits photosynthetic capacity, reducing ITE (Schutz et al., 2009).

The WUE* and IWUE* of Australian sites ranged from 0.5 to 3.5 gC kgH$_2$O$^{-1}$ and 5.6 to 29.8 gC hPa kgH$_2$O$^{-1}$, respectively (Table 3), which is lower than global estimates (Beer et al. (2009). A direct comparison with global results is difficult due to the dissimilar ecosystem types; however, Beer et al. (2009) obtained values of 3.1 gC kgH$_2$O$^{-1}$ and 30.3 gC

hPa kgH$_2$O$^{-1}$ for WUE* and IWUE*, respectively, in the evergreen broadleaf biome type. Across similar C3 broadleaf systems in Australia, both WUE* and IWUE* were lower at 2.2 gC kgH$_2$O$^{-1}$ and 16.7 gC hPa kgH$_2$O$^{-1}$. Previous research using Australian data have shown WUE* to be largest in evergreen broadleaf forest (EBF) sites (~3.0 gC kgH$_2$O$^{-1}$), followed by savanna sites (1.5 gC kgH$_2$O$^{-1}$) and grassland (~0.9 gC kgH$_2$O$^{-1}$) (Shi et al., 2014). They demonstrated the climate dependency of WUE* on VPD and soil water content, hence the preferred use of IWUE* here. Eamus et al. (2013b) examined WUE* and

IWUE* for a tropical Mulga woodland and observed: 1) that daily WUE* declined with increasing soil moisture content in both wet and dry seasons and declined with increasing VPD only in the dry season, a result attributed to an interaction of soil moisture content with VPD in the wet; 2) IWUE* declined with increasing soil moisture content and increased with increasing VPD in both seasons.

# 5 Synergies with modelling and remote sensing

## 5.1 Synergies between remote sensing and the OzFlux network

Satellite derived meteorology and optical spectral vegetation indices (VIs) have been used extensively to scale flux tower datasets in space and time (Chen et al., 2007; Huete et al., 2008; Muraoka and Koizumi, 2008; Verma et al., 2014). Satellite- derived land cover data provide information on spatial variations in vegetation type and structure, which is then used




to interpolate between flux tower sites. Moreover, intra-annual and long term remote sensing products can elucidate the timing

of plant growth / seasonality (Ma et al., 2013b) and extent of prior climate, fire, land use, and disturbances, thus helping better

understand observed fluxes (Asner, 2013; Baumann et al., 2014; Wulder and Franklin, 2006). In some instances, satellite data

have been used to gap fill meteorological data when required for the generation of model drivers and annual budgets (de

Goncalves et al., 2009; Reichle et al., 2011; Restrepo-Coupe et al., 2013). Vegetation indices and other biophysical products

(e.g. MODIS LAI and fPAR) constitute measures of ecosystem structure (e.g. quantity of leaves, (Sea et al., 2011) and function

(e.g. quality of leaves) and represent the phenological drivers of productivity, transpiration and other key ecosystem fluxes

(Restrepo-Coupe et al., 2015; Zhang et al., 2010). Therefore, quantification of surface characteristics via satellite products

improves flux studies and provides a more robust analysis of carbon, energy and water cycles. The integration of eddy

covariance and remote sensing datasets has driven recent efforts to measure optical properties at flux sites, closing the gap

between sampling of temporal and spatial scales (Gamon et al., 2006, 2010).

Conversely, flux data and ancillary *in situ* measurements associated with eddy covariance systems have been

extensively used for the validation of different satellite products (e.g. MODIS GPP and ET (Kanniah et al., 2009b; Restrepo-

Coupe et al., 2015)) and to assist in the parameterisation of models that rely on remotely sensed data (e.g. GPP, ET, canopy

conductance, and light use efficiency (LUE)) (Barraza et al., 2014, 2015; Glenn et al., 2011; Goerner et al., 2011). Given the

challenge of managing water in the dry Australian continent, remote sensing of actual ET is a crucial task and Glenn et al.

(2011) provide an overview of the Australian experience in this task. Similarly, *in situ* fluxes can provide the basic information

required for the interpretation of satellite derived measures of greenness (Huete et al., 2006, 2008; Restrepo-Coupe et al.,

2015). Recently, comparisons between flux data and satellite products have been proposed as a tool to evaluate sensor

continuity, e.g. transition from MODIS derived VIs to the Visible - Infrared Imaging Radiometer Suite (VIIRS) instrument

(Obata et al., 2013). Ground-based flux tower measures, however, offer much more than validation of remote sensing products

and models. An understanding of why satellite – flux tower relationships hold or don't hold can greatly advance and contribute

to our understanding of mechanisms underpinning carbon and water cycles and scaling factors.



### 5.2 Synergies between terrestrial biosphere modelling and the OzFlux network

Development and validation of terrestrial biosphere models are reliant on observational data. Here we refer to examples of the utility of OzFlux data for advancing these models. OzFlux data were instrumental in constraining a continent-wide assessment of terrestrial carbon and water cycles (Haverd et al., 2013a). That paper explored the utility of multiple observation types (streamflow, measurements of evapotranspiration (ET) and net ecosystem production (NEP) from 12 eddy-flux sites, litterfall data and data on carbon pools) to constrain a terrestrial biosphere land surface model of Australian terrestrial

carbon and water fluxes. They conclude that eddy flux measurements provide a significantly tighter constraint on continental net primary production (NPP) than all the other data types. Nonetheless, simultaneous constraint by multiple data types is important for mitigating bias from any single type. Four significant results emerged from the multiply constrained model of the 1990–2011 period: 1) on the Australian continent, a predominantly semi-arid region, over half the water loss through ET ($0.64 \pm 0.05$) occurred through soil evaporation and bypassed plants entirely; 2) mean Australian NPP was quantified at $2.2 \pm 0.4$

PgC yr$^{-1}$, with a significant reduction in uncertainty compared with previous estimates; 3) annually cyclic ("grassy") vegetation and persistent ("woody") vegetation accounted for $0.67 \pm 0.14$ and $0.33 \pm 0.14$ PgC yr$^{-1}$, respectively, of NPP across Australia; 4) the average inter-annual variability of Australia's NEP ($\pm 0.18$ PgC yr$^{-1}$) was larger than Australia's total anthropogenic greenhouse gas emissions in 2011 (0.149 PgC equivalent yr$^{-1}$) and was dominated by variability in semi-arid regions. Results from the above model-data synthesis were used to produce major flux components of the first ever full terrestrial carbon budget

of Australia (Haverd et al., 2013b) as part of a larger international effort to reconcile bottom-up and top-down estimates of the global carbon budget. Further applications include an assessment of the climate sensitivity of Australian carbon and water cycles (Raupach et al., 2013) and an assessment of the magnitude of the Australian contribution to the record global sink anomaly of 2011, with counter-evidence to the assertion by Poulter et al. (2014) that Australian semi-arid ecosystems have entered a regime of increased sensitivity of NEP to precipitation (Haverd et al., 2015).

OzFlux data have also featured in the development of new models. For example, they have been used as constraints and validation for reductionist approaches to modelling evapotranspiration and canopy conductance at continental Australian (Guerschman et al., 2009) and global scales (Yebra et al., 2012). They have also been critical to the development of novel model parameterisations for heat storage in vegetation (Haverd et al., 2007), in-canopy turbulence (Haverd et al., 2009), stable-

isotope transport in soil and vegetation (Haverd and Cuntz, 2010) and canopy radiative transfer (Haverd et al., 2012). More
recently they have been employed as constraints in the development of a novel approach to modelling coupled carbon
allocation and phenology in savanna ecosystems, leading to emergent predictions about the controls of tree cover in Australian
savannas (Haverd et al., 2016). Alternate modelling strategies such as use of stomatal optimality theory has been developed
and tested in Australian savannas using OzFlux data (Schymanski et al., 2008a, 2008b).

## 6 Future outlook

The OzFlux network has been highly successful in generating standardised measurements and protocols, as well as for
providing advanced QA/QC data compatible with international databases (FLUXNET) (Papale et al., 2006) under an open
access data policy. OzFlux has contributed to the FLUXNET community efforts to improve data processing algorithms to
minimise potential errors associated with night-time bias, gap filling and lack of energy balance closure (Baldocchi, 2003; van
Gorsel et al., 2009). This has enabled a significant uptake of the eddy covariance data for application to a range of research
questions as exemplified above. OzFlux is also aligned with the long-term plan for Australian ecosystem science (Andersen
et al., 2014).

Dynamic global models are based on the notion that the same environmental controls will produce the same vegetation
structure irrespective of environmental and evolutionary history (Lehmann et al., 2014). The unique evolutionary history of
the Australian continent, climate and vegetation underpins the importance of the OzFlux network as it provides Australian
derived data for key ecosystem metrics such as NEP, GPP, RUE and IWUE* for use in continental and global vegetation
models. Despite the dominance of the semi-arid/arid climate and the importance of semi-arid ecosystems in Australia and
globally, there still remains a gap in our knowledge about the effect of soil water deficits, soil evaporation, extreme
temperatures and vapour pressure deficits (Eamus et al., 2013b) both as ecosystem drivers and as extreme events that
accompany drought. Currently, ecosystem response to drought is not well understood, particularly because low precipitation
events themselves are unpredictable in timing, duration and severity. It is expected that the frequency and severity of drought
will increase with climate change; therefore, our current understanding of responses to rainfall scarcity will aid in our
understanding of ecological responses to future climate and the potential consequences and adaptation that may be required.

A number of drought related questions can be addressed by OzFlux including: 1) how do droughts affect physiological

processes such as photosynthesis at leaf to landscape scales; 2) what is the effect of seasonal droughts *versus* multi-year

droughts;  3) are there critical thresholds or compensatory reductions in productivity due to drought;  and 4) are there multi-

year drought legacy or feedback effects?

We have demonstrated that OzFlux has already contributed to other areas of importance such as ecosystem responses

to fire, pest outbreaks, cyclones, and the impact of land use and land cover change (including urban flux systems (Coutts et

al., 2007, 2010)) (Baldocchi, 2014a).  Future climate is likely to be more variable and extreme and the network is well placed

to capture and understand these events (Frank et al., 2015).

While there is no significant influence of temperature on NEP or GPP across Australian biomes the strong dependence

of these variables on MAP indicates that even in currently well-watered areas the combined effect of increased temperature

and VPD, similar or reduced water availability will potentially place these systems under stress.  OzFlux then is well positioned

to provide continuous assessment of the long-term condition of these ecosystems and provide early warning across multiple

ecosystems of changes in plant performance as the planet moves into the more forceful climate of the Anthropocene.

There are also opportunities for OzFlux to play a core role in investigating processes within the 'Critical Zone', defined

as the Earth's outer layer from vegetation canopy and through the soil and groundwater that sustains human life (Lin, 2010).

Critical zone science extends us from the thin surface layer to the larger critical zone domain that allows a comparison of the

environmental effects across gradients of climate (Cernusak et al., 2011), time, lithology, human disturbance, biological

activity and topography (Lin, 2010).  This should also involve a focus on greenhouse gas exchanges (not just $CO_2$ but also

$N_2O$ and $CH_4$) from the soil and understanding the linkages with microbial and rhizosphere processes (Hinsinger et al., 2009;

Livesley et al., 2011).  Some of OzFlux contribution here can be to utilise overstorey and understorey flux measurements to

understand the role of distinct understorey vegetation such as the grasses versus trees in savannas (Moore et al., 2015)

Baldocchi (2014b) reminds us that additional ecological and physiological measurements (function, structure, pools

and turnover) add significant value to our flux data and are required for modelling carbon pools.  OzFlux works closely with

the SuperSite Facility (Karan et al., 2016) in TERN as each SuperSite is required to co-host an OzFlux tower. Over the last 4

years ecophysiological data have been collected across 7 sites and along with prior data from a number of OzFlux sites this



has allowed an early assessment of plant performance to be made across many of the OzFlux sites. The SuperSites facility

complements the OzFlux measurements by measuring the biotic community and biophysical environment. Co-location of

activities for both facilities has been important as this has enabled both networks to share resources and survive through tough

financial times. Like any long-term network in Australia this ability to survive is a hall-mark of success (Lindenmayer et al.,

2012). Hence, it is imperative that going forward OzFlux enhances and utilises the synergies and research collaborations

between OzFlux and other TERN platforms such as remote sensing (AusCover), modelling (eMAST) and plot networks such

as transects, AusPlots and the long term ecological research network (LTERN). No single capability is able to address all of

the spatial, ecological, human and cultural challenges required. Instead a synergistic approach (Peters and Loescher, 2014) that

is policy relevant (with political commitment) is required to monitor and manage our planet.

**Acknowledgments**

This work utilised data collected by grants funded by the Australian Research Council (DP0344744, DP0772981,

DP120101735, DP130101566, LE0882936). Beringer is funded under an ARC Future Fellowship (FT110100602). Support

for OzFlux is provided through the Australia Terrestrial Ecosystem Research Network (TERN) (http://www.tern.org.au).

Haverd's contribution was supported by the Australian Climate Change Science Program.

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





**Figure 1: Major Australian biomes defined using the Interim Biogeographic Regionalisation for Australia v. 7 (IBRA) (Environment, 2012). Flux sites from Table 1 are shown illustrating the wide geographical and biome space but each biome is not equally represented. Only a small fraction of the OzFlux sites (8%) are located in the arid/semi-arid biomes that comprises 74% of the landscape.**





**Figure 2: OzFlux sites (white stars) across bioclimatic space with Fluxnet codes (Table 1). Mean annual temperature and mean annual precipitation are shown as the climate space using mean annual (1961-1990) gridded data (0.1 degree**
**resolution)** *(Bureau-of-Meteorology, 2013)*. **The colours represent the major modified Köppen climate classes (Stern and Dahni, 2013) as follows: 1) Equatorial (Pink), Tropical (Dark Blue), Sub-tropical (Light Blue), Desert (Green), Grassland (Yellow) and Temperate (Red).**






**Figure 3: Weekly ensemble of the measured Net Ecosystem Productivity (NEP) (a) and component Gross Primary Production (GPP) (b) and Ecosystem Respiration (ER) (b) for all OzFlux tower sites in each biome as follows** ⬤—⬤ **temperate woodlands,** ▼▼ **tropical savannas,** ▲▲ **desert shrublands,** ◀◀ **tropical grasslands,** ▶▶ **mediterranean woodands,** ⬤—⬤ **temperate broadleaf forests,** ☐—☐ **pasture,** ⬠—⬠ **tropical moist broadleaf forests**

**and** ✦—✦ **montane grasslands. The sites used in each biome are shown in Table 1. GPP generally scales with Leaf Area Index (LAI) and water availability. Most of the variability follows the respective rainfall patterns associated with the tropical, mediterranean, and temperate climates.**





**Figure 4:** **Relationship between site averaged Ecosystem Respiration (ER) and Gross Primary Production (GPP) for all OzFlux tower sites (blue squares) showing they are well correlated ($r^2$=0.93) with a slope of 0.76. This compares with slope of 0.77 in an international synthesis by Baldocchi (2008). The results from the international study (red and green symbols) are shown in the background and show a secondary line where disturbed sites (green triangles) had large positive values of Net Ecosystem Productivity (NEP) (carbon source). Few Australian sites reach that level of disturbance. Solid lines are regressions, dashed lines are 95 % confidence intervals**






**Figure 5: Simple heat map of annual average OzFlux tower measurements to identify the correlations between fluxes, driving variables and ecosystem indices using all available site years of data. Fluxes are gross primary productivity (GPP), ecosystem**
**respiration (ER), net ecosystem production (NEP), evapotranspiration (ET). Drivers are leaf area index (LAI), precipitation (Precip), air temperature ($T_a$), incoming solar radiation ($F_{sd}$) and vapor pressure deficit (VPD). We used the following ecosystem indices: radiation use efficiency (RUE) following Garbulsky et al. (2010); ecosystem water-use-efficiency (WUE\*) and inherent ecosystem WUE\* (IWUE\*) following Beer et al. (2009); Bowen ratio (BR) is defined as the ratio between sensible and latent heat fluxes (Bowen, 1926); Leaf area index (LAI) was obtained from the average of the MODIS MOD15 product for the site years**
**available that has been de-spiked using the procedure described in Kanniah et al. (2009a); and rainfall use efficiency (Rain Use Eff) was defined following Huxman et al. (2004) as the ratio of GPP to precipitation. GPP and RUE are also normalized by LAI by dividing them by LAI to produce series such as GPP/LAI. Colour scale indicates the strength of Pearson correlation co-efficient (r).**





*Figure 6: The relationships of OzFlux tower data by ecoregion type (Table 1) between gross primary production (GPP) and*
*the major climate drivers (a) mean annual precipitation (MAP) and (b) mean annual temperature (MAT). Also shown is*
*the relationship between GPP and (c) actual evapotranspiration (AET) and (d) radiation use efficiency (RUE). The global*
*relationships in Garbulsky et al. (2010) are shown in the background to aid comparison. Simple curve fits are shown to*
*aid visualization.*





**Table 1: A summary of the OzFlux tower sites along with the available data range and site history. The associated IBRA vegetation class is given (IBRA) along with the broader world ecoregion classification, both from the Interim Biogeographic Regionalisation for Australia v. 7 (IBRA) (Environment, 2012).**

| Site name | FLUXNET ID | Date range | Latitude S | Longitude E | Elevation (m) | IBRA | World ecoregion | Site history |
|---|---|---|---|---|---|---|---|---|
| Adelaide River | AU-Ade | 2007-2009 | -13.08 | 131.12 | 100 | PCK | Tropical savannas | Woodland savanna on shallow loamy sand (hydrolsol) with low level grazing pressure, fire frequency 1 in 3 years. |
| Alice Springs | AU-ASM | 2010-present | -22.28 | 133.25 | 606 | BRT | Deserts and Xeric Shrublands | Pine Hill cattle station. Cattle grazing in low numbers has continued for decades. |
| Arcturus | AU-Arc | 2011-2013 | -23.86 | 148.47 | 178 | BBN | Tropical grassland | Pasture (to W) Lightly forested tussock grasslands: Crop (to E): 2011-June 2012: Barren land due to flooding, followed by remediation work, June 2012-November 2012: Chickpeas, November 2012-June 2013: Fallow, June 2013-November 2013: Wheat, November 2013-January 2014: Fallow |





| Calperum | AU-Cpr | 2010-present | -34.00 | 140.59 | 53 | MDD | Mediterranean woodands | Multi-stemmed, 3-5m high eucalypt trees centred 2-4m apart with limited mid-story species, extensive perennial spinifex and occasional annual grasses. Regrowth of trees after fire ~ 35years ago with sheep grazing ceased in 1994. Site burned by wildfire in January 2014 and regrowth now occurring. |
|---|---|---|---|---|---|---|---|---|
| Cape Tribulation | AU-Ctr | 2001-present | -16.10 | 145.45 | 66 | WET | Tropical and Sub-tropical moist broadleaf | Selective logging finished. World Heritage, National Park fetch region. |
| Cow Bay | AU-Cow | 2009-present | -16.24 | 145.43 | 86 | WET | Tropical and Sub-tropical moist broadleaf forests | Selective logging finished. World Heritage, National Park fetch region. |
| Cumberland Plains | AU-Cum | 2012-present | -33.62 | 150.72 | 54 | SYB | Temperate woodlands | Not been managed but is a remnant woodland at the urban-wildland interface. |





| Daly River Pasture | AU-DaP | 2007-2013 | -14.13 | 131.39 | 67 | DAB | Tropical pasture | Tropical improved pasture ($C_4$ grass and legume mixture), cleared 1982, low grazing pressure (~1 head ha$^{-1}$), fire 1 in 5 years. |
|---|---|---|---|---|---|---|---|---|
| Daly River Savanna | AU-DaS | 2007-present | -14.16 | 131.39 | 53 | DAB | Tropical savannas | Woodland savanna on deep sandy loam soils (kandosol), native pasture grazing, very low grazing pressure (~0.1 head ha$^{-1}$), fire 1 in 5 years. |
| Dargo | AU-Drg | 2007-present | -37.13 | 147.17 | 1648 | AUA | Montane grasslands | Low density historical and current grazing during the warmer months . |
| Dry River | AU-Dry | 2008-present | -15.26 | 132.37 | 180 | STU | Tropical savannas | Woodland savanna on sandy loam soils, native pasture grazing, low grazing pressure (~0.5 head ha$^{-1}$), fire 1 in 8 years. |
| Fogg Dam | AU-Fog | 2006-2008 | -12.55 | 131.31 | 4 | DAC | Tropical wetland | Seasonally inundated wetland, conservation reserve, heavy clay soils (vertosol). |





| Gingin | AU-Gin | 2011-present | -31.38 | 115.71 | 53 | SWA | Mediterranean woodands | The only current active (and accidental) management is fire, with the Dept of Parks and Wildlife conducting controlled fuel reduction burns approximately every 20 years. The area immediately to the north was burned in a wildfire in 2006, and to the south has not been burned for at least 30 years. |
| Great Western Woodland | AU-GWW | 2013-present | -30.19 | 120.65 | 486 | COO | Mediterranean woodands | Managed as conservation reserve since 2007; sheep and cattle grazing from 1906 to 2007 led to some soil surface degradation. |
| Howard Springs | AU-How | 2001-present | -12.49 | 131.15 | 41 | DAC | Tropical savannas | Open-forest savanna on sandy loam soils, peri-urban water catchment reserve, previously grazed (1960s), fire frequency 1 in 2 years . |



| Litchfield | AU-Lit | 2015-present | -13.18 | 130.79 | | DAC | Tropical savannas | Open-forest savanna on sandy loam soils, national park, fire frequency 1 in 2 years. |
|---|---|---|---|---|---|---|---|---|
| Nimmo | AU-Nim | 2007-present | -36.22 | 148.55 | 1337 | AUA | Montane grasslands | Low density historical and current grazing during the warmer months. |
| Otway | AU-Otw | | -38.53 | 142.81 | | SCP | Pasture | |
| Red Dirt Melon Farm | AU-RDF | 2011-2013 | -14.56 | 132.48 | 171 | DAB | Tropical savannas | Woodland savanna on sandy loam soils, cleared and converted to cropland in 2012, fire frequency (in uncleared state) 1 in 12 years. |
| Ridgefield | AU-Rgf | 2015-present | -32.51 | 116.97 | | AVW | Pasture | |
| Riggs Creek | AU-Rig | 2010-present | -36.65 | 145.58 | 162 | RIV | Pasture | Ongoing cattle and sheep grazing. |
| Robson Creek | AU-Rob | 2013-present | -17.12 | 145.63 | 710 | WET | Tropical and Sub-tropical moist broadleaf forests | Selective logging finished 1969. World Heritage, National Park fetch region. Traditional use of the forest by Tablelands Yidinji had finished by the 1960s. |




| Samford | Au-Sam | 2010- present | -27.39 | 152.88 | 87 | SEQ | Pasture | Native woodland clearly over 100 years ago, grazed tropical native and improved grasses supporting subsistence livestock production until 2009. Simulated grazing from 2009. |
| Sturt Plains | AU-Stp | 2008- present | -17.15 | 133.35 | 225 | MGD | Tropical grasslands | Mitchel grassland plain on cracking clay soils (vertosol), very low grazing pressure (<0.1 head ha$^{-1}$), fire 1 in 16 years. |
| Ti Tree East | AU-TTE | 2012- present | -22.29 | 133.64 | 553 | BRT | Deserts and Xeric Shrublands | Pine Hill cattle station. Cattle grazing ceased between 2009 – 2013 but was re-introduced in in July 2014 and stocked continuously in low-to-moderate densities (~3000 head of cattle on ca. 700 km$^2$). |
| Tumbarumba | AU-Tum | 2001- present | -35.66 | 148.15 | 1249 | NSS | Temperate Broadleaf and mixed Forest | 2003 insect attack, 2004 selective logging in footprint. |
| Virginia Park | AU-Vir | | -19.88 | 146.55 | | EIU | Savanna | |





| Wallaby Creek | AU-Wac | 2005-2013 | -37.43 | 145.19 | 738 | SHE | Temperate Broadleaf and mixed Forest | Old growth forest burned by stand replacing fire in Feb 2009 and now regrowing forest. |
| Warra | Au-Wrr | 2013-present | -43.10 | 146.65 | 121 | TSR | Temperate Broadleaf and mixed Forest | Natural forest that has originated from natural disturbance by period wildfire (most recent wildfire was in 1898). |
| Whroo | AU-Whr | 2011-present | -36.67 | 145.03 | 143 | VIM | Temperate woodlands | Clearing in the broader area during the 19th century to supply timber for gold mining. It is also thought that there was selective logging in the area during much of the 20th century. Some of this is most likely relatively recent, with numerous tree stumps present at the site. |
| Wombat | AU-Wom | 2010-present | -37.42 | 144.09 | 715 | VIM | Temperate broadleaf forest | Site is a ~ 25 years old secondary regrowth forest with a tree height of 21–25 m. Forest |



| | | | | | | | | management includes rotational prescribed burns of understorey vegetation and shelter wood harvesting. |
|---|---|---|---|---|---|---|---|---|
| Yanco | AU-Ync | 2012-present | -34.99 | 146.29 | 128 | RIV | Temperate Grasslands | Ongoing cattle and sheep grazing. |





**Table 2: Summary of the representation of OzFlux tower sites within each ecoregion compared with the total percentage of the continent comprising this ecoregion (Environment, 2012). The mean carbon fluxes are given for each ecoregion type.**

| Ecoregion | Percentage of the continent comprising this ecoregion (%) | Percentage of flux towers in that ecoregion (%) | GPP (tC ha$^{-1}$ yr$^{-1}$) | NEP (tC ha$^{-1}$ yr$^{-1}$) | ER (tC ha$^{-1}$ yr$^{-1}$) |
|---|---|---|---|---|---|
| Tropical and subtropical moist broadleaf forests | <1 | 12 | 22.1 | 2.8 | 19.3 |
| Temperate broadleaf and mixed forest | 7 | 16 | 21.5 | 3.9 | 17.6 |
| Tropical and subtropical grasslands, savannas and shrublands | 30 | 28 | 14.1 | 1.7 | 12.4 |
| Temperate grasslands, savannas and shrublands | 3 | 16 | 14.5 | 3.4 | 11.1 |
| Montane grasslands and shrublands | <1 | 8 | 10.6 | 1.2 | 9.4 |
| Mediterranean forests, woodlands and scrub | 11 | 12 | 6.7 | 0.2 | 6.5 |
| Deserts and xeric shrublands | 49 | 8 | 1.8 | -1.1 | 2.8 |





Table 3: Site averaged data for each OzFlux tower for available data periods (Table 1) showing mean and standard deviation of the daily fluxes. Fluxes are gross primary productivity (GPP), ecosystem respiration (ER), net ecosystem production (NEP), evapotranspiration (ET). Drivers are leaf area index (LAI), precipitation (Precip), air temperature (Ta), incoming solar radiation (Fsd) and vapour pressure deficit (VPD). We used the following ecosystem indices: radiation use efficiency (RUE) following Garbulsky et al. (2010); ecosystem(*) water-use-efficiency (WUE*) and inherent ecosystem WUE* (IWUE*) following Beer et al. (2009); Bowen ratio (BR) is defined as the ratio between sensible and latent heat fluxes (Bowen, 1926); Leaf area index (LAI) was obtained from the average of the MODIS MOD15 product for the site years available that has been de-spiked using the procedure described in Kanniah et al. (2009a); and rainfall use efficiency (Rain Use Eff) was defined following Huxman et al. (2004) as the ratio of GPP to precipitation. Some quantities are also normalized by LAI by dividing them by dividing by LAI to produce series such as GPP/LAI.

| Site | Data years inclusive | GPP tC ha$^{-1}$ yr$^{-1}$ | GPP/LAI tC ha$^{-1}$ yr$^{-1}$ m$^2$ m$^{-2}$ | ER tC ha$^{-1}$ yr$^{-1}$ | NEP tC ha$^{-1}$ yr$^{-1}$ | Fsd MJ m$^{-2}$ d$^{-1}$ | Fn MJ m$^{-2}$ d$^{-1}$ | LAI m$^2$ m$^{-2}$ | VPD kPa | Ta °C | Ah g m$^{-3}$ | Precip mm yr$^{-1}$ | WUE* gC kgH$_2$O | IWUE* gC kgH$_2$O | IWUE*_LAI gC kgH$_2$O hPa m$^2$ m$^{-2}$ | RUE gC MJ PAR$^{-1}$ | Rain Use Eff tC ha$^{-1}$ yr$^{-1}$ mm$^{-1}$ | BR | ET mm yr$^{-1}$ |
|---|---|---|---|---|---|---|---|---|---|---|---|---|---|---|---|---|---|---|---|
| Adelaide River | 2007-2009 | 18.8 ± 10.5 | 15.9 ± 5.2 | 15.3 ± 5.0 | 3.5 ± 2.3 | 20.5 ± 2.5 | 12.2 ± 0.4 | 1.1 ± 0.5 | 1.4 ± 1.7 | 26.9 ± 4.7 | 16.2 | 1472.7 ± 1612.3 | 1.6 ± 0.4 | 21.1 ± 8.2 | 21.8 ± 12.9 | 1.2 | 12.8 | 0.7 ± 0.6 | 1120.8 ± 457.4 |
| Alice Springs | 2010-2014 | 3.2 ± 3.5 | 8.6 ± 5.9 | 3.5 ± 3.1 | -0.3 ± 1.3 | 22.6 ± 3.7 | 11.2 ± 3.0 | 0.3 ± 0.1 | 2.1 ± 0.8 | 22.8 ± 5.7 | 6.9 ± 2.8 | 331.3 ± 451.3 | 1.9 ± 1.7 | 29.8 ± 18.9 | 108.8 ± 89.3 | 0.4 | 9.6 | 15.1 ± 21.4 | 263.9 ± 272.5 |
| Arcturus | 2011-2013 | 4.7 ± 3.1 | 6.6 ± 3.7 | 5.1 ± 1.3 | -0.4 ± 2.7 | 19.6 ± 4.2 | 10.5 ± 3.2 | 0.7 ± 0.3 | 1.3 ± 0.5 | 21.6 ± 4.7 | 10.7 ± 2.8 | 638.8 ± 575.2 | 1.2 ± 0.6 | 13.8 ± 6.9 | 20.2 ± 8.3 | 0.4 | 7.3 | 2.1 ± 1.4 | 398.9 ± 165.7 |
| Calperum | 2010-2014 | 5.2 ± 2.3 | 10.5 ± 3.1 | 4.6 ± 1.5 | 0.6 ± 1.5 | 18.4 ± 6.9 | 8.7 ± 4.6 | 0.5 ± 0.1 | 1.3 ± 0.7 | 17.9 ± 5.3 | 7.5 ± 1.3 | 300.6 ± 312.8 | 1.7 ± 0.7 | 20.7 ± 10.9 | 44.2 ± 21.6 | 0.6 | 17.1 | 2.9 ± 2.0 | 300.4 ± 139.9 |
| Cow Bay | 2009-2014 | 20.5 ± 1.4 | 4.6 ± 1.1 | 18.1 ± 2.0 | 2.4 ± 2.3 | 12.1 ± 2.9 | 7.3 ± 2.2 | 4.7 ± 0.9 | 0.6 ± 0.1 | 23.6 ± 2.2 | 17.4 ± 2.5 | 3930.0 ± 4631.5 | 1.9 ± 0.4 | 11.0 ± 4.5 | 2.4 ± 1.1 | 1.3 | 5.2 | 0.1 ± 0.2 | 1087.1 ± 251.7 |
| Cumberland | 2013-2014 | 9.1 ± 3.1 | 7.4 ± 2.1 | 9.8 ± 3.5 | -0.7 ± 3.0 | 12.6 ± 8.8 | 7.2 ± 5.3 | 1.3 ± 0.2 | 0.8 ± 0.3 | 18.1 ± 4.1 | 10.6 ± 2.8 | 806.3 ± 787.3 | 1.5 ± 0.7 | 11.7 ± 3.9 | 9.0 ± 3.5 | 0.4 | 9.7 | 0.7 ± 7.3 | 550.1 ± 273.1 |
| Daly River Pasture | 2007-2013 | 14.0 ± 13.5 | 6.9 ± 5.5 | 12.9 ± 8.1 | 1.1 ± 6.3 | 21.3 ± 2.1 | 10.9 ± 1.8 | 1.5 ± 0.8 | 1.4 ± 0.5 | 25.3 ± 2.9 | 14.6 ± 4.6 | 1237.8 ± 1701.0 | 1.5 ± 0.7 | 18.0 ± 6.6 | 13.7 ± 8.3 | 0.6 | 11.3 | 3.4 ± 4.0 | 740.7 ± 580.3 |
| Daly River Uncleared | 2007-2014 | 14.3 ± 6.7 | 11.6 ± 2.9 | 13.9 ± 5.7 | 0.5 ± 2.7 | 20.8 ± 2.3 | 11.8 ± 2.7 | 1.2 ± 0.3 | 1.6 ± 0.5 | 26.6 ± 2.5 | 14.5 ± 4.4 | 1197.5 ± 1550.0 | 1.6 ± 0.3 | 24.5 ± 8.4 | 22.2 ± 10.4 | 0.9 | 12 | 0.9 ± 0.5 | 869.6 ± 297.9 |



| Site | Years | | | | | | | | | | | | | | | | | | |
|---|---|---|---|---|---|---|---|---|---|---|---|---|---|---|---|---|---|---|---|
| Dargo | 2007-2014 | 9.5 ± 5.8 | 4.3 ± 1.5 | 7.8 ± 4.1 | 1.8 ± 2.0 | 16.6 ± 6.6 | 7.6 ± 5.5 | 2.1 ± 0.9 | 0.3 ± 0.2 | 6.8 ± 5.0 | 5.7 ± 1.3 | 509.2 ± 420.3 | 1.6 ± 0.4 | 5.6 ± 3.5 | 2.6 ± 1.1 | 0.5 | 18.7 | 0.3 ± 0.6 | 551.4 ± 331.8 |
| Dry River | 2008-2014 | 12.3 ± 4.9 | 10.3 ± 2.6 | -11.4 ± 3.7 | 0.9 ± 2.4 | 22.0 ± 2.3 | 11.8 ± 2.5 | 1.2 ± 0.2 | 1.8 ± 0.6 | 26.8 ± 3.4 | 13.7 ± 4.6 | 836.4 ± 1093.2 | 1.6 ± 0.4 | 28.3 ± 13.2 | 26.7 ± 16.1 | 0.7 | 14.7 | 1.2 ± 1.1 | 805.4 ± 361.5 |
| Fogg Dam | 2006-2008 | 9.5 ± 3.8 | 4.2 ± 1.3 | 6.3 ± 0.8 | 3.2 ± 3.4 | 21.3 ± 2.4 | 13.4 ± 1.8 | 2.2 ± 0.3 | 1.0 ± 0.3 | 25.6 ± 2.4 | 17.1 ± 3.8 | 1440.2 ± 1650.0 | 0.7 ± 0.2 | 6.5 ± 1.2 | 3.0 ± 0.7 | 0.4 | 6.6 | 0.3 ± 0.3 | 1355.3 ± 260.6 |
| Gingin | 2011-2014 | 11.2 ± 3.2 | 12.8 ± 2.3 | 10.9 ± 1.5 | 0.3 ± 2.2 | 19.8 ± 7.0 | 11.0 ± 4.9 | 0.9 ± 0.2 | 1.0 ± 0.6 | 18.7 ± 4.9 | 10.2 ± 1.4 | 546.7 ± 479.5 | 1.9 ± 0.5 | 20.5 ± 14.5 | 22.0 ± 13.4 | 0.9 | 20.5 | 1.5 ± 1.3 | 601.1 ± 184.8 |
| Great Western Woodland | 2013-2014 | 3.9 ± 0.8 | 10.2 ± 1.7 | 4.1 ± 1.5 | -0.2 ± 1.1 | 20.0 ± 6.3 | 11.4 ± 5.0 | 0.4 ± 0.1 | 1.5 ± 0.7 | 19.9 ± 5.2 | 7.8 ± 1.5 | 361.1 ± 430.1 | 1.6 ± 0.4 | 21.6 ± 9.1 | 55.7 ± 19.5 | 0.5 | 10.7 | 3.0 ± 1.3 | 261.7 ± 105.9 |
| Howard Springs | 2001-2014 | 20.5 ± 7.3 | 13.3 ± 2.7 | 16.3 ± 5.2 | 4.2 ± 3.3 | 20.2 ± 2.2 | 13.2 ± 2.4 | 1.5 ± 0.4 | 1.2 ± 0.4 | 26.7 ± 1.8 | 16.9 ± 3.8 | 1468.6 ± 1890.1 | 1.7 ± 0.3 | 21.3 ± 5.9 | 15.7 ± 8.0 | 1.1 | 14 | 0.6 ± 0.5 | 1135.1 ± 350.2 |
| Nimmo | 2007-2014 | 11.7 ± 7.3 | 6.6 ± 2.4 | 11.0 ± 5.9 | 0.7 ± 2.7 | 17.2 ± 6.9 | 9.0 ± 5.3 | 1.6 ± 0.5 | 0.4 ± 0.2 | 8.0 ± 4.9 | 6.1 ± 1.4 | 620.5 ± 443.2 | 1.8 ± 0.5 | 6.8 ± 3.9 | 3.9 ± 1.5 | 0.6 | 18.8 | 0.4 ± 0.3 | 637.0 ± 366.5 |
| Riggs Creek | 2010-2014 | 10.5 ± 8.0 | 8.1 ± 4.3 | 9.2 ± 3.0 | 1.3 ± 5.6 | 17.5 ± 7.5 | 8.2 ± 4.4 | 1.3 ± 0.9 | 0.9 ± 0.6 | 15.6 ± 5.4 | 8.1 ± 1.3 | 395.8 ± 397.6 | 2.2 ± 1.5 | 12.5 ± 9.8 | 17.5 ± 35.1 | 0.7 | 26.6 | 1.3 ± 1.7 | 462.7 ± 228.9 |
| Robson Creek | 2013-2015 | 23.7 ± 3.2 | 5.8 ± 1.3 | 20.6 ± 3.8 | 3.2 ± 2.3 | 18.2 ± 4.8 | 12.7 ± 3.3 | 4.3 ± 1.1 | 0.4 ± 0.1 | 20.1 ± 2.8 | 14.7 ± 2.4 | 2199.4 ± 2420.8 | 2.0 ± 0.4 | 8.7 ± 4.1 | 2.0 ± 0.7 | 1 | 10.8 | 0.4 ± 0.3 | 1185.8 ± 223.6 |
| Samford | 2010-2014 | 16.6 ± 7.2 | 9.3 ± 3.3 | 14.3 ± 5.1 | 2.2 ± 3.3 | 17.0 ± 4.2 | 9.9 ± 3.3 | 1.7 ± 0.3 | 0.9 ± 0.3 | 19.4 ± 3.9 | 11.3 ± 4.0 | 1240.6 ± 1430.2 | 2.8 ± 0.5 | 24.1 ± 10.0 | 14.3 ± 6.9 | 0.9 | 13.3 | 0.9 ± 0.5 | 564.2 ± 215.7 |
| Sturt Plains | 2008-2014 | 4.8 ± 5.0 | 7.5 ± 7.4 | 4.6 ± 2.5 | 0.2 ± 2.9 | 22.7 ± 2.4 | 9.9 ± 2.4 | 0.5 ± 0.2 | 2.0 ± 0.7 | 26.0 ± 3.9 | 11.1 ± 4.6 | 730.5 ± 1172.5 | 0.8 ± 0.5 | 13.1 ± 7.5 | 26.5 ± 15.9 | 0.4 | 6.5 | 3.8 ± 4.4 | 549.6 ± 427.4 |
| Ti Tree East | 2012-2014 | 0.4 ± 2.7 | 0.4 ± 5.9 | 2.2 ± 2.2 | -1.8 ± 1.2 | 22.5 ± 3.9 | 9.6 ± 2.7 | 0.3 ± 0.1 | 2.4 ± 0.9 | 23.8 ± 6.0 | 6.1 ± 2.5 | 259.9 ± 468.2 | 0.5 ± 0.3 | 11.7 ± 8.1 | 30.3 ± 10.6 | 0 | 1.4 | 19.2 ± 23.0 | 214.0 ± 275.4 |
| Tumbarumba | 2001-2014 | 22.1 ± 7.7 | 5.3 ± 1.7 | 17.0 ± 6.5 | 5.1 ± 3.7 | 16.6 ± 7.0 | 8.7 ± 5.3 | 4.1 ± 0.6 | 0.5 ± 0.3 | 9.6 ± 5.3 | 6.4 ± 1.4 | 1924.2 ± 1400.4 | 2.6 ± 0.5 | 11.7 ± 7.6 | 2.8 ± 1.7 | 1 | 11.5 | 0.6 ± 0.3 | 850.7 ± 361.7 |



| Site | Period | | | | | | | | | | | | | | | | | | |
|---|---|---|---|---|---|---|---|---|---|---|---|---|---|---|---|---|---|---|
| Wallaby Creek | 2005-2013 | 19.8 ± 5.4 | 5.2 ± 1.2 | 10.4 ± 5.9 | 9.3 ± 6.7 | 12.1 ± 5.3 | 7.5 ± 5.3 | 3.9 ± 0.8 | 0.4 ± 0.3 | 10.8 ± 4.0 | 7.2 ± 1.3 | 1571.2 ± 1098.1 | 2.4 ± 0.9 | 9.9 ± 5.9 | 2.5 ± 1.4 | 1.3 | 12.6 | 0.3 ± 0.4 | 911.3 ± 472.2 |
| Warra | 2013-2014 | 22.1 ± 8.8 | 13.8 ± 2.0 | 23.8 ± 2.5 | -1.7 ± 6.8 | 10.4 ± 6.1 | 2.5 ± 12.2 | 1.6 ± 0.7 | 0.4 ± 0.2 | 10.0 ± 2.7 | 7.1 ± 0.6 | 1053.0 ± 459.8 | 3.5 ± 0.7 | 11.9 ± 4.9 | 7.4 ± 1.3 | 2.8 | 21 | -0.1 ± 0.6 | 643.2 ± 325.5 |
| Whroo | 2011-2014 | 13.0 ± 3.0 | 14.1 ± 2.8 | 10.0 ± 2.0 | 3.0 ± 2.2 | 17.7 ± 7.9 | 9.7 ± 5.8 | 0.9 ± 0.1 | 0.9 ± 0.5 | 16.0 ± 4.9 | 8.1 ± 1.2 | 381.3 ± 329.5 | 2.6 ± 0.5 | 23.2 ± 14.6 | 25.0 ± 15.2 | 1.1 | 34.2 | 1.4 ± 0.9 | 494.9 ± 157.7 |
| Wombat | 2010-2014 | 22.5 ± 7.3 | 5.5 ± 1.4 | 13.5 ± 4.6 | 9.1 ± 3.9 | 15.4 ± 7.3 | 10.4 ± 5.8 | 4.1 ± 0.7 | 0.4 ± 0.3 | 11.4 ± 4.1 | 8.0 ± 1.3 | 924.9 ± 567.4 | 3.2 ± 0.6 | 12.1 ± 8.5 | 3.0 ± 2.2 | 1.1 | 24.3 | 0.3 ± 0.5 | 724.2 ± 331.9 |
| Yanco_JAXA | 2012-2014 | 4.8 ± 3.6 | 8.0 ± 4.0 | 4.7 ± 1.7 | 0.1 ± 2.2 | 19.0 ± 7.7 | 9.5 ± 4.8 | 0.5 ± 0.3 | 1.3 ± 0.8 | 17.3 ± 6.2 | 7.1 ± 1.0 | 472.1 ± 374.6 | 1.9 ± 1.2 | 16.2 ± 9.1 | 37.9 ± 37.2 | 0.5 | 10.1 | 4.0 ± 4.5 | 266.5 ± 135.3 |