# Peer review of "An introduction to the Australian and New Zealand flux tower network – OzFlux"

_Biogeosciences, 2016_

## Referee Comment (RC1) · D. Papale (Referee) · 20 May 2016

Beringer and co-authors submitted a nice overview and synthesis of the OzFlux network where its importance is well documented and explained. For the nature and content of the paper however I think that it should be submitted as "Review and synthesis" and not "Research article". The main limitation in the paper that I think must be addressed is that OzFlux includes both Australia and New Zealand (as also cited by the authors) but the paper is only on Australian sites. This is an important limitation. I can understand that there could be data policy limitations and issues however at least in the general part of the description the New Zealand sites could be added and described. Clearly it would be better if the sites can be added also to the analysis.

Other comments:

[Figure]

- Affiliation 23, the word "Australia" is missing

- Affiliation 25 is not an affiliation

- It would be important to stress the unique characteristics (in general) of the Australian sites respect to the rest of FLUXNET and why their contribution is crucial globally.

- Line 100: other more recent examples exist respect to Running 1999, I suggest to add them also to better highlight the role of eddy covariance measurements in recent activities.

- Line 110: you forgot Europe, that is an historical network that together with AmeriFlux were the start of the FLUXNET...

- Lines 167-168: it refers to standard protocols that however should be better explained at least with references. The same is valid for the list of common measurements present at all the sites: it would be an important info to add.

- Lines 174-178: it is not clear which are the tasks specific of the central hub respect to the site managers. Who is doing the quality control and processing – postprocessing?

- In the Section 5.2 it could be added the importance of the OzFlux sites also in the empirical upscaling models since they are covering unique climate and vegetation.

- I suggest to add the significance of the correlation in figure 5. In addition the use of MODIS LAI should be better evaluated. It is not a measurement and has uncertainty: why not using measured LAI? Or if satellite data are needed why not Vegetation Indexes (direct measurements)?

- Still on Figure 5, there are mixed the interannual variability and spatial variability. This makes the analysis difficult to interpret because in the network there are sites with long time series and sites with only few years of data. A possible solution could be to redo (or add) the figure using only average multi-year data (analyzing only spatial variability).

- The table 1 is quite large and difficult to read and use. I suggest a more condensed

version and the rest probably as supplementary data, better if directly as csv file. I suggest also to report the coordinates of the sites with at least 4 decimal digits: these data will be used by people working in remote sensing and it is important to give the best information available.

---

## Referee Comment (RC2) · Anonymous Referee #2 · 30 May 2016

The manuscript provides detailed introduction of the background of OzFlux network and its evolution. Synthetic description for various Australian sites is also informative. However, first of all, there is a discrepancy between the title (Australian and New Zealand flux tower network) and the contents (data only from Australian network). I would recommend changing the title to fit the contents such as "the Australian flux tower network". More desirably, the authors could modify discussions significantly including the data from sites located in New Zealand. Otherwise the main discussion would not match the title emphasizing "Australian and New Zealand flux tower network".

Secondly, I would like to encourage the authors to include more target-oriented data analyses. Figures 4-6 express static relationships between meteorological elements and carbon- and water-cycle components, and that would be important basic information. However, the authors could expand their analyses based on their matter of interest expressed in the introduction, for example, risks of fire, disease, management practices and land-use changes under future climate change. Technological advances have also accelerated during the past several years for better regional estimates based on both up-scaled flux-tower data-sets and inversion analyses (I suppose that CSIRO is one of the leading institutes on the topic). How well the Australian tower flux data would contribute to solve some of such issues or improve estimates of regional and global carbon- and water-cycles? I suggest that the authors would show some more new hypothesis or attempts in the manuscript in order to answer such questions using the Australian data-sets. At present, the manuscript includes sufficient information of introduction of Australian flux sites and reviews of their studies, however, original scientific findings are relatively limited.

Specific comments: Pages 4-5 "The role of flux research in Australia": I would recommend referring more papers from Oceania and stating more region-specific issues. In particular, more recent papers would have been published for topics listed in Page 5, lines 93-100.

Pages 12-15 "Biotic and abiotic controls on land-surface exchanges": This part well describes specific characteristics of Australian surface processes. However, the contents was relatively limited in the past studies and showing only Fig.5. More original and new scientific questions would be desirable to be discussed. For example; were there any long-term trends detected in spatial distribution of each flux, biomass, LAI, species composition, etc.? Since the OzFlux seemed to have a good collaboration with TERN as described in "Introduction", I would expect that the data and understandings obtained by TERN could be used for interpretation of long-term trends in biotic and abiotic controls on land-surface exchanges.

---

## Author Comment (AC1) · 30 Aug 2016

**Response to reviewers comments on "An introduction to the Australian and New Zealand flux tower network – OzFlux" by Jason Beringer et al.**

**Comment by D. Papale (Referee) darpap@unitus.it**

Beringer and co-authors submitted a nice overview and synthesis of the OzFlux network where its importance is well documented and explained. For the nature and content of the paper however I think that it should be submitted as "Review and synthesis" and not "Research article". The main limitation in the paper that I think must be addressed is that OzFlux includes both Australia and New Zealand (as also cited by the authors) but the paper is only on Australian sites. This is an important limitation. I can understand that there could be data policy limitations and issues however at least in the general part of the description the New Zealand sites could be added and described. Clearly it would be better if the sites can be added also to the analysis.

- We will work with our colleagues to include all available long term sites in NZ to the data tables and analysis. We will include additional information on the history of the NZ science as well as differences between the two countries within the OzFlux network in terms of climate, instrumentation, etc.. We anticipate to redraw figures as necessary.

Other comments:

- Affiliation 23, the word "Australia" is missing
  - Will correct
- Affiliation 25 is not an affiliation
  - Will remove
- It would be important to stress the unique characteristics (in general) of the Australian sites respect to the rest of FLUXNET and why their contribution is crucial globally.
  - The reviewer raises a good point here so we will take the opportunity to add the following text, "Australian vegetation is quite dissimilar to the northern Hemisphere as a result of continental isolation, tectonic-geological history and climate that results in the dominance of sclerophyllous, evergreen, woody species that do not fit in well with the common global plant functional types as discussed in Peel et al. (2005). The ancient nutrient poor soils of Australia have driven the evolution of woody and sclerophyllous vegetation such as eucalyptus and acacias which are characterized by the presence of small, rigid, long-lived leaves (Peel et al., 2005). The eucalypts and acacias are predominately evergreen broadleaf plant functional types and represent a large fraction of this type globally as represented in global climate models. The OzFlux network is the only source of flux information for the eucalypts and acacias that dominate the continent and, as these life forms are not significantly represented in natural biomes outside Australia, OzFlux is of considerable importance to FluxNet in completely the global picture. These vegetation groups occur primarily in arid and semiarid climates that dominate the Australian landscape (this paper) and provide a crucial source of information in understanding the role of semiarid vegetation in the global carbon cycle (Ahlström et al., 2015; Poulter et al., 2014).

- Line 100: other more recent examples exist respect to Running 1999, I suggest to add them also to better highlight the role of eddy covariance measurements in recent activities.
  - A good suggestion. We will add a number of references covering the breadth of the role of EC in remote sensing better. i.e. (Anav et al., 2015; Moore et al., 2016; Running et al., 1999; Schimel et al., 2015).

- Line 110: you forgot Europe, that is an historical network that together with AmeriFlux were the start of the FLUXNET. . .
  - We didn't intend to make an exhaustive list here but will add one of the most important ones, the European flux network.
- Lines 167-168: it refers to standard protocols that however should be better explained at least with references. The same is valid for the list of common measurements present at all the sites: it would be an important info to add.
  - We refer the reader at this point in the text to the section describing this in the paper documenting the network setup and modify the text as "OzFlux established an agreed set of core measurements and common protocols for measurements of carbon, water and energy fluxes across the national network (see section 3) to provide consistent observations to serve the land surface and ecosystem modelling communities."

- Lines 174-178: it is not clear which are the tasks specific of the central hub respect to the site managers. Who is doing the quality control and processing – postprocessing?
  - We suggest modifying the text to "The central node implements a centralised database and provides feedback on live data feeds (equipment failure) and measurement quality to site operators. Individual site operators have responsibility for tower operation, data processing and quality control using OzFluxQC, gap filling, post-processing and are then required to deliver data streams to the central database."
- In the Section 5.2 it could be added the importance of the OzFlux sites also in the empirical upscaling models since they are covering unique climate and vegetation.
  - This is another good point and we will add the following "Finally, flux tower information for Australia has been used in empirical upscaling methods (such as machine learning) that uses gridded satellite information and meteorology to produce global estimates of carbon and water budgets (Jung et al., 2009, 2011). These utilised some of the earlier data from Howard Springs, Tumbarumba and Wallaby Creek in the la Thuile fluxnet database that helped constrain the global uncertainties."
- I suggest to add the significance of the correlation in figure 5. In addition the use of MODIS LAI should be better evaluated. It is not a measurement and has uncertainty: why not using measured LAI? Or if satellite data are needed why not Vegetation Indexes (direct measurements)?
  - It is not currently straight forward to add the significance to the plot. If this is important then it can be done and we leave this to the editors discretion. Yes we point this out in the paper "Note that the magnitude of LAI from the MODIS LAI product utilised in this paper varies from site based estimates but has been used for consistency." We make a stronger case for its use in the text as below "Here we use MODIS LAI purely in a relative sense to assess the relative differences in cover and how they may influence the observed fluxes. Many sites have no LAI measurements and some others have ad-hoc measurements over time. In addition, we know that the magnitude of LAI from the MODIS LAI product utilised in this paper varies from site based estimates but has been used for consistency. ".
- Still on Figure 5, there are mixed the interannual variability and spatial variability. This makes the analysis difficult to interpret because in the network there are sites with long time series and sites with only few years of data. A possible solution could be to redo (or add) the figure using only average multi-year data (analyzing only spatial variability).
  - The review makes a good point about mixing spatial and temporal variability and we were concerned about this. We checked the data used it is was in fact site averaged data so the figure just includes spatial variability. We will keep the figure as is but amend the caption to read " Simple heat map of Australian OzFlux tower measurements to identify the correlations between fluxes, driving variables and ecosystem indices using all site averaged data for available site years to represent spatial variability."

- The table 1 is quite large and difficult to read and use. I suggest a more condensed version and the rest probably as supplementary data, better if directly as csv file. I suggest also to report the coordinates of the sites with at least 4 decimal digits: these data will be used by people working in remote sensing and it is important to give the best information available.
  - Will submit the site history information as a supplement.
  - Good idea. We will add further decimal places to the coordinates

**Anonymous Referee #2**

The manuscript provides detailed introduction of the background of OzFlux network and its evolution. Synthetic description for various Australian sites is also informative. However, first of all, there is a discrepancy between the title (Australian and New Zealand flux tower network) and the contents (data only from Australian network). I would recommend changing the title to fit the contents such as "the Australian flux tower network". More desirably, the authors could modify discussions significantly including the data from sites located in New Zealand. Otherwise the main discussion would not match the title emphasizing "Australian and New Zealand flux tower network".

- This comment is similar to reviewer 1 and we address that above

Secondly, I would like to encourage the authors to include more target-oriented data analyses. Figures 4-6 express static relationships between meteorological elements and carbon- and water-cycle components, and that would be important basic information. However, the authors could expand their analyses based on their matter of interest expressed in the introduction, for example, risks of fire, disease, management practices and land-use changes under future climate change. Technological advances have also accelerated during the past several years for better regional estimates based on both up-scaled flux-tower data-sets and inversion analyses (I suppose that CSIRO is one of the leading institutes on the topic). How well the Australian tower flux data would contribute to solve some of such issues or improve estimates of regional and global carbon- and water-cycles? I suggest that the authors would show some more new hypothesis or attempts in the manuscript in order to answer such questions using the Australian data-sets. At present, the manuscript includes sufficient information of introduction of Australian flux sites and reviews of their studies, however, original scientific findings are relatively limited.

- Clearly thematic and systematic issues based analyses are crucial in advancing the science and our understanding of Australian ecosystems. However, this is an overview of the network and specific analyses are beyond the scope of this paper. However, these are outlined as future directions and opportunities for the network and will be the focus of OzFlux science in the coming decade. We also already discuss the utility of the flux information for regional modelling and remote-sensing. Moreover, there are a number of papers in the special issue that address these explicitly. We will make better connections to these in the paper.

Specific comments: Pages 4-5 "The role of flux research in Australia": I would recommend referring more papers from Oceania and stating more region-specific issues. In particular, more recent papers would have been published for topics listed in Page 5, lines 93-100.

- More recent references will be given as below. This section is however addressing how EC as a method can address, however, we will rewrite this list as follows:

  1) providing accurate, continuous half-hourly to annual estimates of sinks and sources of greenhouse gases and water from ecosystems for carbon accounting and water management that is particularly important in such an arid country as Australia (Hutley et al., 2005; Raupach et al., 2013);

2) evaluating the effects of disturbance, topography, biodiversity, stand age, insect/pathogen infestation and extreme weather on carbon and water fluxes particularly cyclone, fire and heat waves in the Australian environment (Beringer et al., 2014; Bowman et al., 2009; van Gorsel et al., 2016; Hutley et al., 2013);

3) examining the effects of land management practices, such as harvest, fertilisation, irrigation, tillage, thinning, cultivation and clearing, especially agriculture in the region (Bristow et al., 2016; Campbell et al., 2015; Rutledge et al., 2015); and

4) producing important ground-truth data for parameterising, validating, and improving satellite remote sensing and global inversion products (Anav et al., 2015; Moore et al., 2016; Running et al., 1999; Schimel et al., 2015), particularly phenology (Ma et al., 2013; Moore et al., 2016) and water balance.

Pages 12-15 "Biotic and abiotic controls on land-surface exchanges": This part well describes specific characteristics of Australian surface processes. However, the contents was relatively limited in the past studies and showing only Fig.5. More original and new scientific questions would be desirable to be discussed. For example; were there any long-term trends detected in spatial distribution of each flux, biomass, LAI, species composition, etc.? Since the OzFlux seemed to have a good collaboration with TERN as described in "Introduction", I would expect that the data and understandings obtained by TERN could be used for interpretation of long-term trends in biotic and abiotic controls on land-surface exchanges.

- We feel that these are best discussed in the section on "Future outlook" and we already have questions posed on drought response and disturbance. The reviewer highlights an important point about long term trends and currently the network is limited in long term sites that have enough years to attribute temporal changes. This is a good suggestion from the reviewer so we will add the following paragraph to the Future outlook section.

[revised manuscript text omitted]

---

## Author Response (AR2)

**Response to reviewers comments on "An introduction to the Australian and New Zealand flux tower network – OzFlux" by Jason Beringer et al.**

In addition to the response to the reviewer comments we have also made the following changes as kindly suggested by the editor:

1. L114: Have removed "ChinaFlux".
2. L183-187: Have rephrased not to be questions according to the comments that "The authors raised three key questions, but it is unclear where they responded. I recommend answering to those questions at the last chapter with bullet points. If the authors did not intend to answer those questions, I would recommend removing this as readers might be confused. I found other three questions at the last chapter (L522-525); although I am a big fan of questions, I felt too many questions without answers in a manuscript would make readers empty. "
3. L130: Have removed one dot after "records"
4. L369: Have added "In determining RUE we calculate the absorbed PAR using the MODIS fraction of absorbed PAR product (MODIS MOD15 fPAR) and flux tower PAR data." in response to "Please explain how APAR was computed to estimate RUE. It seems Garbulsky et al., (2010) used MODIS fPAR, thus I guess the authors used MODIS fPAR and flux tower PAR data. "
5. L374: Have removed "Rain Use Eff".

**Comment by D. Papale (Referee) darpap@unitus.it**

Beringer and co-authors submitted a nice overview and synthesis of the OzFlux network where its importance is well documented and explained. For the nature and content of the paper however I think that it should be submitted as "Review and synthesis" and not "Research article". The main limitation in the paper that I think must be addressed is that OzFlux includes both Australia and New Zealand (as also cited by the authors) but the paper is only on Australian sites. This is an important limitation. I can understand that there could be data policy limitations and issues however at least in the general part of the description the New Zealand sites could be added and described. Clearly it would be better if the sites can be added also to the analysis.

- We have worked with our colleagues to include all available long term sites in NZ to the data tables and analysis (this was in the end only 2 long term sites Troughton Farm and Kopuatai). We have included additional information on the history of the NZ science as well as differences between the two countries within the OzFlux network in terms of climate, instrumentation, etc.. Including the following comments into the manuscript.
    - The two New Zealand sites represented extremes in productivity for a moist temperate climate zone with the grazed dairy farm site having the highest GPP of any OzFlux site (2620 gC m$^{-2}$ yr$^{-1}$) and the natural raised peat bog site had very low GPP (820 gC m$^{-2}$ yr$^{-1}$).
    - Although New Zealand (NZ) flux sites have been an integral part of OzFlux from the outset and have made many important contributions (Campbell et al., 2014, 2015; Hunt et al., 2002, 2016; Rutledge et al., 2010, 2015) these sites have had a different history with typically shorter site records from primarily managed systems. Thus this paper will largely focus on Australian sites, with the addition of two of the NZ sites with longer multi-year records.
    - New Zealand has a different history of flux sites, with long-term sites being slower to become established because of the shorter-term nature of the funding system. With much of the New Zealand economy centred on the agricultural sector, and efforts to ensure their sustainability and reduced greenhouse gas emissions, there has been a recent strong research focus on mitigation of soil carbon losses, including the use of EC techniques. Having experienced a large rate of native biodiversity loss, there have also been EC studies carried out in indigenous ecosystems, including tussock grasslands, forests, and wetlands.

- o Most NZ sites have developed separate data processing protocols and systems compared to the Australian part of OzFlux, with uptake of EddyPro post-processing becoming more standard in the last few years.
  - o NZ's climate is more temperate maritime with a wide latitudinal range, but generally younger landscapes and precipitation is generally evenly distributed except for strong precipitation gradients associated with mountain chains in both main islands.
  - o New Zealand sites have more variable instrument systems, with earlier measurements made using closed path gas analysers (LI-6262, LI-7000 LI-COR, Lincoln, Nebraska, USA), but now with open path and "enclosed" path sensors used (LI-7500, LI-7200, LI-COR, Lincoln, Nebraska, USA).
  - o Data from NZ sites were gap-filled and partitioned using advanced neural networks (following Papale and Valentini (2003) as implemented in Matlab (The Mathworks, Natick, Massachusetts, USA).
  - o One of the major impacts on New Zealand ecosystem carbon and water exchanges occurs as a result of seasonal drought. For grazed pasture, Rutledge et al. (2015) and Mudge et al. (2011) showed that NEP of a dairy farm during a year with a severe drought largely recovered to pre-drought levels over the remainder of the year because of the year-round growing conditions. In a raised peat bog, Goodrich et al. (2015a) found that GPP was reduced under conditions of elevated VPD common during drought, and Goodrich et al. (2015b) described reductions in methane fluxes for up to six months after water tables recovered following drought.
  - o
- • We have reanalysed and redrawn figures 2 and 4.
- • Data has been added to tables 1 and 2.

Other comments:

- • Affiliation 23, the word "Australia" is missing
  - o Have corrected
- • Affiliation 25 is not an affiliation
  - o Have removed
- • It would be important to stress the unique characteristics (in general) of the Australian sites respect to the rest of FLUXNET and why their contribution is crucial globally.
  - o The reviewer raises a good point here so we have taken the opportunity to add the following text, "Australian vegetation is quite dissimilar to the northern Hemisphere as a result of continental isolation, tectonic-geological history and climate that results in the dominance of sclerophyllous, evergreen, woody species that do not fit into global plant functional types as discussed in Peel et al. (2005). Australian nutrient poor soils drive woody and sclerophyllous vegetation that are characterized by the presence of small, rigid, long-lived leaves (Peel et al., 2005). Importantly the OzFlux regional network is the only source of flux information for eucalypts and acacias that dominate the continent and are not significantly represented outside Australia. These vegetation groups occur primarily in arid and semiarid climates that dominate the landscape (this paper) and provide a crucial source of information in understanding the role of Australian semiarid vegetation in the global carbon cycle (Ahlström et al., 2015; Poulter et al., 2014). The eucalypts and acacias are predominately evergreen broadleaf plant functional types and represent a large fraction of this type globally that is represented in global climate models and information from this regional network would enable explicit ecological and physiological characteristics as well as the behaviour of Eucalyptus for future climate modelling."

- • Line 100: other more recent examples exist respect to Running 1999, I suggest to add them also to better highlight the role of eddy covariance measurements in recent activities.

- o A good suggestion. We have added a number of references covering the breadth of the role of EC in remote sensing better. i.e. (Anav et al., 2015; Moore et al., 2016; Running et al., 1999; Schimel et al., 2015).

- Line 110: you forgot Europe, that is an historical network that together with AmeriFlux were the start of the FLUXNET. . .
  - o We didn't intend to make an exhaustive list here but we have added one of the most important ones, the European flux network.
- Lines 167-168: it refers to standard protocols that however should be better explained at least with references. The same is valid for the list of common measurements present at all the sites: it would be an important info to add.
  - o We refer the reader at this point in the text to the section describing this in the paper documenting the network setup and modify the text as "OzFlux established an agreed set of core measurements and common protocols for measurements of carbon, water and energy fluxes across the national network (see section 3) to provide consistent observations to serve the land surface and ecosystem modelling communities."

- Lines 174-178: it is not clear which are the tasks specific of the central hub respect to the site managers. Who is doing the quality control and processing – postprocessing?
  - o We have modified the text to "The central node implements a centralised database and provides feedback on live data feeds (equipment failure) and measurement quality to site operators. Individual site operators have responsibility for tower operation, data processing and quality control using OzFluxQC, gap filling, post-processing and are then required to deliver data streams to the central database."
- In the Section 5.2 it could be added the importance of the OzFlux sites also in the empirical upscaling models since they are covering unique climate and vegetation.
  - o This is another good point and have added the following "Finally, flux tower information for Australia has been used in empirical upscaling methods (such as machine learning) that uses gridded satellite information and meteorology to produce global estimates of carbon and water budgets (Jung et al., 2009, 2011). These utilised some of the earlier data from Howard Springs, Tumbarumba and Wallaby Creek in the la Thuile fluxnet database that helped constrain the global uncertainties."
- I suggest to add the significance of the correlation in figure 5. In addition the use of MODIS LAI should be better evaluated. It is not a measurement and has uncertainty: why not using measured LAI? Or if satellite data are needed why not Vegetation Indexes (direct measurements)?
  - o It is not currently straight forward to add the significance to the plot. If this is important then it can be done and we leave this to the editors discretion. Yes we point this out in the paper "Note that the magnitude of LAI from the MODIS LAI product utilised in this paper varies from site based estimates but has been used for consistency." We now make a stronger case for its use in the text as below "Here we use MODIS LAI purely in a relative sense to assess the relative differences in cover and how they may influence the observed fluxes. Many sites have no LAI measurements and some others have ad-hoc measurements over time. In addition, we know that the magnitude of LAI from the MODIS LAI product utilised in this paper varies from site based estimates but has been used for consistency. ".
- Still on Figure 5, there are mixed the interannual variability and spatial variability. This makes the analysis difficult to interpret because in the network there are sites with long time series and sites with only few years of data. A possible solution could be to redo (or add) the figure using only average multi-year data (analyzing only spatial variability).
  - o The reviewer makes a good point about mixing spatial and temporal variability and we were concerned about this. We checked the data used it is was in fact site averaged data so the figure

just includes spatial variability. We will keep the figure as is but have amended the caption to read " Simple heat map of Australian OzFlux tower measurements to identify the correlations between fluxes, driving variables and ecosystem indices using all site averaged data for available site years to represent spatial variability."

- The table 1 is quite large and difficult to read and use. I suggest a more condensed version and the rest probably as supplementary data, better if directly as csv file. I suggest also to report the coordinates of the sites with at least 4 decimal digits: these data will be used by people working in remote sensing and it is important to give the best information available.
    - Will submit the site history information as a supplement.
    - Good idea. We have added further decimal places to the coordinates

**Anonymous Referee #2**

The manuscript provides detailed introduction of the background of OzFlux network and its evolution. Synthetic description for various Australian sites is also informative. However, first of all, there is a discrepancy between the title (Australian and New Zealand flux tower network) and the contents (data only from Australian network). I would recommend changing the title to fit the contents such as "the Australian flux tower network". More desirably, the authors could modify discussions significantly including the data from sites located in New Zealand. Otherwise the main discussion would not match the title emphasizing "Australian and New Zealand flux tower network".

- This comment is similar to reviewer1 and we have addressed that above

Secondly, I would like to encourage the authors to include more target-oriented data analyses. Figures 4-6 express static relationships between meteorological elements and carbon- and water-cycle components, and that would be important basic information. However, the authors could expand their analyses based on their matter of interest expressed in the introduction, for example, risks of fire, disease, management practices and land-use changes under future climate change. Technological advances have also accelerated during the past several years for better regional estimates based on both up-scaled flux-tower data-sets and inversion analyses (I suppose that CSIRO is one of the leading institutes on the topic). How well the Australian tower flux data would contribute to solve some of such issues or improve estimates of regional and global carbon- and water-cycles? I suggest that the authors would show some more new hypothesis or attempts in the manuscript in order to answer such questions using the Australian data-sets. At present, the manuscript includes sufficient information of introduction of Australian flux sites and reviews of their studies, however, original scientific findings are relatively limited.

- Clearly thematic and systematic issues based analysis are crucial in advancing the science and our understanding of Australian ecosystems. However, this is an overview of the network and specific analysis are beyond the scope of this paper. However, these are outlined as future directions and opportunities for the network and will be the focus of OzFlux science in the coming decade. We also already discuss the utility of the flux information for regional modelling and remote-sensing. Moreover, there are a number of papers in the special issue that address these explicitly. We have made better connections to these in the paper.

Specific comments: Pages 4-5 "The role of flux research in Australia": I would recommend referring more papers from Oceania and stating more region-specific issues. In particular, more recent papers would have been published for topics listed in Page 5, lines 93-100.

- More recent references are included in the text as described below. This section is however addressing how EC as a method can address, however, we have rewritten this list as follows:

1) providing accurate, continuous half-hourly to annual estimates of sinks and sources of greenhouse gases and water from ecosystems for carbon accounting and water management that is particularly important in such an arid country as Australia (Hutley et al., 2005; Raupach et al., 2013);

2) evaluating the effects of disturbance, topography, biodiversity, stand age, insect/pathogen infestation and extreme weather on carbon and water fluxes particularly cyclone, fire and heat waves in the Australian environment (Beringer et al., 2014; Bowman et al., 2009; van Gorsel et al., 2016; Hutley et al., 2013);

3) examining the effects of land management practices, such as harvest, fertilisation, irrigation, tillage, thinning, cultivation and clearing, especially agriculture in the region (Bristow et al., 2016; Campbell et al., 2015; Rutledge et al., 2015); and

4) producing important ground-truth data for parameterising, validating, and improving satellite remote sensing and global inversion products (Anav et al., 2015; Moore et al., 2016; Running et al., 1999; Schimel et al., 2015), particularly phenology (Ma et al., 2013; Moore et al., 2016) and water balance.

Pages 12-15 "Biotic and abiotic controls on land-surface exchanges": This part well describes specific characteristics of Australian surface processes. However, the contents was relatively limited in the past studies and showing only Fig.5. More original and new scientific questions would be desirable to be discussed. For example; were there any long-term trends detected in spatial distribution of each flux, biomass, LAI, species composition, etc.? Since the OzFlux seemed to have a good collaboration with TERN as described in "Introduction", I would expect that the data and understandings obtained by TERN could be used for interpretation of long-term trends in biotic and abiotic controls on land-surface exchanges.

- We feel that these are best discussed in the section on "Future outlook" and we already have questions posed on drought response and disturbance. The reviewer highlights an important point about long term trends and currently the network is limited in long term sites that have enough years to attribute temporal changes. This is a good suggestion from the reviewer so we have added the following paragraph to the Future outlook section.

[revised manuscript text omitted]